# CrossPan: A Comprehensive Benchmark for Cross-Sequence Pancreas MRI Segmentation and Generalization

**Linkai Peng**[*1,2]                LINKAI.PENG@NORTHWESTERN.EDU
**Cuiling Sun**[*1,3]           CUILINGSUN2026@U.NORTHWESTERN.EDU
**Zheyuan Zhang**[*1]         ZHEYUANZHANG2023@U.NORTHWESTERN.EDU
**Wanying Dou**[1]               MONICA.DOU@NORTHWESTERN.EDU
**Halil Ertugrul Aktas**[1]        HALILERTUGRUL.AKTAS@NORTHWESTERN.EDU
**Andrea M Bejar**[1]             ANDREA.BEJAR@NORTHWESTERN.EDU
**Elif Keles**[1]                ELIF.KELES@NORTHWESTERN.EDU
**Tamas Gonda**[4]              TAMAS.GONDA@NYULANGONE.ORG
**Michael B Wallace**[5]            WALLACE.MICHAEL@MAYO.EDU
**Zongwei Zhou**[6]                  ZZHOU82@JH.EDU
**Gorkem Durak**[1]             GORKEM.DURAK@NORTHWESTERN.EDU
**Rajesh N Keswani**[7]           RAJ-KESWANI@NORTHWESTERN.EDU
**Ulas Bagci**[1]               ULAS.BAGCI@NORTHWESTERN.EDU

[1] *Department of Radiology, Northwestern University, Chicago, USA.*

[2] *Department of Electrical and Computer Engineering, Northwestern University, Evanston, USA.*

[3] *Department of Computer Science, Northwestern University, Evanston, USA.*

[4] *Department of Gastroenterology, New York University, NYC, USA.*

[5] *Division of Gastroenterology and Hepatology, Mayo Clinic in Florida, Jacksonville, USA.*

[6] *Department of Computer Science, Johns Hopkins University, Baltimore, USA.*

[7] *Departments of Gastroenterology and Hepatology, Northwestern University, Chicago, USA.*

**Editors:** Accepted for publication at MIDL 2026

## Abstract

Automatic pancreas segmentation is fundamental to abdominal MRI analysis, yet deep learning models trained on one MRI sequence often fail catastrophically when applied to another—a challenge that has received little systematic investigation. We introduce *CrossPan*, a multi-institutional benchmark comprising 1,386 3D scans across three routinely acquired sequences (T1-weighted, T2-weighted, and Out-of-Phase) from eight centers. Our experiments reveal three key findings. First, cross-sequence domain shifts are far more severe than cross-center variability: models achieving Dice scores above 0.85 in-domain collapse to near-zero (<0.02) when transferred across sequences. Second, state-of-the-art domain generalization methods provide negligible benefit under these physics-driven contrast inversions, whereas foundation models like MedSAM2 maintain moderate zero-shot performance through contrast-invariant shape priors. Third, semi-supervised learning offers gains only under stable intensity distributions and becomes unstable on sequences with high intra-organ variability. These results establish cross-sequence generalization—not model architecture or center diversity—as the primary barrier to clinically deployable pancreas MRI segmentation. Dataset and code are available at https://crosspan.netlify.app/.

**Keywords:** Pancreas Segmentation, MRI, Domain Generalization, Domain Shift, Robustness Evaluation.

---

* Contributed equally

## 1. Introduction

Deep learning has transformed medical image segmentation, yet a critical failure mode remains underexplored: models trained on one MRI sequence often collapse entirely when applied to another. In abdominal imaging, clinicians routinely acquire multiple sequences—T1-weighted (T1W), T2-weighted (T2W), and Out-of-Phase (OOP)—because each reveals distinct tissue properties. T1W emphasizes vascular enhancement, T2W highlights fluid and edema, and OOP suppresses fat signals. This multisequence protocol is clinically essential (Luna et al., 2016; Yoon et al., 2016), but it creates a computational challenge: appearance statistics that are highly predictive in one sequence become inverted or non-informative in another.

Despite its clinical importance, cross-sequence robustness has received little attention in the medical imaging literature. Most prior work on domain generalization focuses on cross-center or cross-scanner variability, implicitly treating MRI as a single homogeneous modality (Safdari et al., 2025). Yet in practice, sequence availability varies substantially across institutions and clinical workflows—patients may undergo only a subset of available protocols depending on their specific indication (Canellas et al., 2019). A segmentation model that performs well on T1W training data but fails on T2W inputs cannot be reliably deployed in such heterogeneous environments.

Pancreas segmentation presents an ideal testbed for studying this problem. The organ's small size, variable shape, and low soft-tissue contrast make it inherently difficult to segment (Zhang et al., 2025), amplifying any degradation caused by domain shift. Moreover, unlike brain or cardiac MRI—where acquisition protocols are relatively standardized—abdominal imaging exhibits substantial inter-institutional variability, making sequence heterogeneity the norm rather than the exception. If cross-sequence robustness cannot be achieved for this challenging anatomy, the prospects for generalized abdominal MRI segmentation are limited.

To address this gap, we introduce **CrossPan**, the first benchmark specifically designed to evaluate cross-sequence generalization in pancreas MRI segmentation. The dataset comprises 1,386 annotated 3D volumes spanning three clinically routine sequences (T1W, T2W, and Out-of-Phase) collected from eight international institutions. To the best of our knowledge, it is the first large-scale, multi-institutional dataset explicitly curated to study the physics-driven domain shifts across T1W, T2W, and OOP sequences. This scale and diversity enable controlled experiments that disentangle sequence-induced shifts from center-induced variability.

We organize our evaluation around five complementary experimental settings: (1) in-domain baselines to establish upper-bound performance, (2) single-source cross-sequence transfer to quantify zero-shot degradation, (3) multi-sequence joint training to measure the achievable ceiling with full data access, (4) leave-one-sequence-out protocols to assess robustness under partial diversity, and (5) semi-supervised learning to evaluate performance under annotation scarcity. Together, these settings isolate the contributions of sequence variability, training data diversity, and supervision level.

Evaluating 15+ methods spanning supervised architectures, domain generalization algorithms, semi-supervised approaches, and foundation models, we find:

- **Cross-sequence shifts dwarf cross-center variability.** Models achieving Dice > 0.85 in-domain collapse to < 0.02 when transferred between sequences—a two-order-of-magnitude degradation that cross-center transfer does not produce.

- **Domain generalization methods fail under physics-driven contrast inversion.** Statistical alignment techniques designed for style or scanner shifts provide no meaningful improvement when the underlying image formation physics changes.

- **Foundation models exhibit divergent behavior.** MedSAM2 maintains moderate zero-shot accuracy through contrast-invariant shape priors, whereas SAM-Med3D requires fine-tuning to function.

- **Semi-supervised learning is sequence-dependent.** Consistency-based SSL improves performance on T1W but becomes unstable on T2W and OOP, where pseudo-label noise compounds across epochs.

These findings establish sequence heterogeneity—not model architecture, center diversity, or annotation volume—as the primary barrier to clinically deployable pancreas MRI segmentation. Code and data are available at https://crosspan.netlify.app/.

## 2. Related Work

### 2.1. Pancreas Segmentation in CT and MRI

Pancreas segmentation has been extensively studied in CT (Roth et al., 2015; Zhou et al., 2017), where standardized Hounsfield Units facilitate model development. In MRI, however, pancreas segmentation remains far less explored due to data scarcity, heterogeneous protocols, and strong sequence-dependent intensity variations. Prior MRI-based studies have mainly focused on single-sequence settings or task-specific applications (e.g., tumor localization or radiotherapy planning), leaving the broader impact of sequence heterogeneity insufficiently examined.

### 2.2. Robust Segmentation Under Domain Shift

Robustness under distribution shift has been investigated primarily in cross-center or cross-scanner scenarios. Large-scale benchmarks such as the M&Ms cardiac challenge (Campello et al., 2021) and prostate MRI datasets (Liu et al., 2020) have motivated numerous domain generalization approaches, including data augmentation (Zhang et al., 2020), feature alignment (Zhou et al., 2021), and invariant representation learning (Krueger et al., 2021). These methods typically assume style- or scanner-driven shifts and are seldom evaluated across heterogeneous MRI sequences. Meanwhile, emerging foundation models and semi-supervised techniques demonstrate strong performance in low-data regimes, but their stability under sequence-level intensity and contrast changes remains largely unverified.

### 2.3. Multi-Sequence MRI & Modality Variability

Multi-sequence modeling is well established in neuroimaging (Menze et al., 2014) and cardiac MRI (Campello et al., 2021), where standardized acquisition enables matched multi-modal protocols. In abdominal MRI, however, acquisition workflows vary widely across

institutions, and patients frequently undergo only a subset of available sequences. As a result, abdominal MRI presents substantial inter-sequence variability, complicating model deployment across heterogeneous protocols. While existing abdominal MRI studies often aggregate sequences into a single domain, systematic evaluation of sequence-specific effects and cross-sequence generalization remains limited.

## 3. Dataset Construction

In this Institutional Review Board (IRB) approved work, we introduce **CrossPan**, a novel and challenging MRI pancreas segmentation dataset designed to systematically analyze domain shift and evaluate domain generalization algorithms. Our cohort included MRI images for pancreas organ segmentation and pancreatic cyst malignancy risk classification from patients referred for evaluation of pancreatic cystic lesions or suspected pancreatic adenocarcinomas. Images were collected from patients 18 years or older who received abdominal MR imaging between March 2004 and June 2024 across seven international centers and one private consortium. These include New York University Langone Health (NYU), Mayo Clinic Florida (MCF), Mayo Clinic Arizona (MCA), Northwestern University (NU), Allegheny Health Network (AHN), Istanbul University Faculty of Medicine (IU), Erasmus Medical Center (EMC), and private In-House consortium data (IH). Select sequences from the raw image files were converted from Digital Imaging and Communications in Medicine (DICOM) to Neuroimaging Informatics Technology Initiative (NIfTI) format prior to collection at each center. Conversion to NIfTI format removes all metadata from images, including all Protected Health Information (PHI), ensuring HIPAA-compliant de-identification.

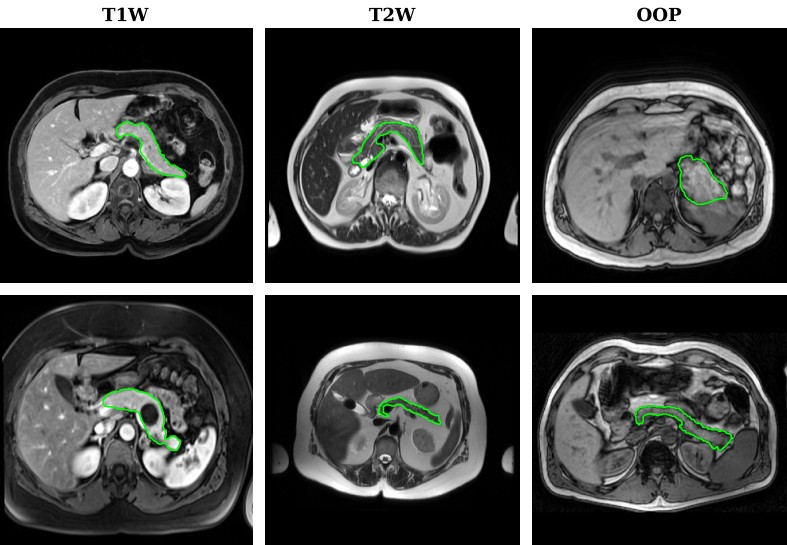

Figure 1: Representative cases across sequences. Example slices from the T1W, T2W, and OOP MRI sequences, with ground-truth pancreas masks shown in green.

**CrossPan** captures three distinct MR sequences: (1) 463 contrast-enhanced T1W, (2) 737 T2W, and (3) 186 OOP. Images were acquired on Siemens, Philips, or GE scanners with either 1.5 T or 3 T field strength. Each sequence exhibits fundamentally different contrast mechanisms, offering a controlled yet realistic setting for isolating sequence-level domain

shifts. Appendix A provides a summary of our dataset. This current dataset is a significant and systematic extension of our prior work (Zhang et al., 2025), with significantly more T1W (385 to 463) and T2W (382 to 737) images and a newly curated OOP (186) sequence.

Figure 1 illustrates representative slices from the three sequences. T1W emphasizes vascular enhancement, T2W scans highlight fluid-tissue interfaces, and OOP imaging suppresses fat signals. These differences lead to pronounced changes in the visibility and sharpness of pancreas boundaries.

To quantify sequence heterogeneity, we visualize the distribution of raw intensities using a 2D t-SNE embedding (Figure 2 (a)). The three sequences form distinct clusters, confirming that global appearance statistics diverge significantly across contrast settings. Even without anatomical features, low-level voxel statistics are sufficient to separate sequences, indicating a strong domain shift.

Beyond whole-volume appearance, we also examine pancreas-specific intensity profiles (Figure 2 (b)). The T2W pancreas exhibits a brighter and narrower histogram due to long-T2 tissues, whereas the OOP pancreas shows suppressed fat signal and reduced dynamic range. T1W scans occupy a mid-intensity distribution. These distinctions validate that the pancreas itself undergoes substantial contrast-dependent variations.

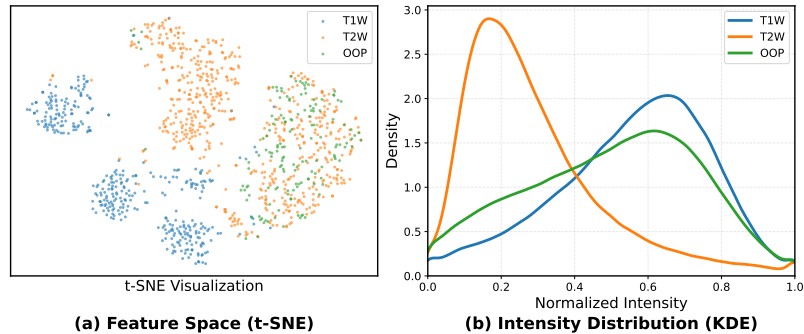

(a) Feature Space (t-SNE)  (b) Intensity Distribution (KDE)

Figure 2: t-SNE embedding of raw MRI volumes colored by sequence. A 2D t-SNE projection of whole-volume intensity features reveals three well-separated clusters corresponding to T1W, T2W, and OOP scans.

## 4. Benchmark Tasks and Setups

We design five evaluation settings to isolate model capacity, data diversity, and sequence-induced domain shifts.

**(1) In-Domain Baseline.** Models are trained and tested on the same sequence to establish upper-bound performance without domain shift.

**(2) Single-Source Cross-Sequence Generalization.** Models trained on a single sequence (e.g., T1W) are directly tested on unseen sequences (e.g., T2W, OOP) to assess zero-shot robustness.

**(3) Multi-Sequence Joint Training.** All sequences are pooled for training to quantify the achievable gap relative to an "ideal" data-rich setting.

**(4) Leave-One-Sequence-Out (LOSO).** Models are trained on two sequences and evaluated on the held-out one to measure robustness under partial sequence diversity.

**(5) Semi-Supervised Learning.** We additionally study robustness under limited annotations (5–40%) using representative SSL techniques.

We evaluate a broad spectrum of segmentation paradigms spanning classical architectures, DG methods, SSL approaches, and foundation models. Details of the models evaluated and training protocols can be found in Appendix B and Appendix C.

We report the Dice Similarity Coefficient (Dice) for volumetric overlap and the 95% Hausdorff Distance (HD95) and Normalized Surface Dice (NSD) to capture geometric boundary precision.

## 5. Results

### 5.1. In-Domain Baseline

We first assess intrinsic model capacity when no sequence shift is present. Table 1 summarizes the in-domain results, while full metrics are provided in Appendix D. For foundation models, we report the zero-shot performances. Performances of other configurations are shown in Table 5.

Across all sequences, established supervised 3D architectures consistently achieve strong Dice performance, indicating that model capacity is not the limiting factor for this benchmark. Among the sequences, T2W MRI yields the highest accuracy, whereas the OOP sequence remains the most challenging due to reduced pancreas–fat contrast.

Domain generalization variants trained in-domain perform similarly to the nnU-Net baseline, confirming that these DG techniques do not adversely affect performance when no domain shift is present.

Interestingly, foundation models show divergent behavior. MedSAM2 achieves competitive zero-shot segmentation across all sequences, reflecting strong global anatomical priors learned during large-scale pretraining. In contrast, SAM-Med 3D variants and Totalsegmentator fail in zero-shot mode and require fine-tuning to produce valid pancreas masks. Even after fine-tuning, SAM-Med 3D variants in-domain accuracy remains lower than specialized biomedical architectures as in Table 5, suggesting that these models struggle with fine-scale pancreatic boundaries despite matched training–testing conditions.

### 5.2. Single-Source Cross-Sequence Generalization

This section evaluates the realistic scenario where models trained on a single sequence are directly applied to a different MRI protocol. Table 2 reports the T1W $\rightarrow$ T2W results, while other cross-sequence directions are provided in Appendix E.

Across all supervised architectures, the performance collapse is dramatic: models trained on T1W achieve Dice scores below 0.02 when tested on T2W. This failure reflects severe overfitting to sequence-specific appearance cues (e.g., dark pancreas vs. bright fat in T1W), which invert under T2W imaging physics. DG algorithms also fail to mitigate this degradation, indicating that statistical alignment does not address contrast-inversion shifts.

To assess whether DG methods provide statistically significant improvements over nnU-Net, we conducted paired Wilcoxon signed-rank tests across cases for each training–testing direction. In single-source settings, performance gains from DG methods are highly sporadic and task-dependent. This lack of consistent statistical significance further validates that style-based or feature-alignment-based DG methods do not provide a reliable solution for the structural contrast changes inherent in abdominal MRI sequence shifts.

Table 1: In-domain Dice performance across T1W, T2W, and OOP MRI sequences. Foundation models are reported in zero-shot mode to highlight their intrinsic transferability, whereas other models are trained on CrossPan.

| Model | T1W Dice | T2W Dice | OOP Dice |
|---|---|---|---|
| **General supervised segmentation models** | | | |
| 3D U-Net | $0.6606 \pm 0.2901$ | $0.8078 \pm 0.0997$ | $0.6729 \pm 0.1596$ |
| SegResNet | $0.6870 \pm 0.2974$ | $0.8421 \pm 0.0755$ | $0.7338 \pm 0.1517$ |
| SwinUNETR | $0.6685 \pm 0.2921$ | $0.8168 \pm 0.0998$ | $0.7040 \pm 0.1645$ |
| SegMamba | $0.8325 \pm 0.0760$ | $0.8298 \pm 0.0875$ | $0.7305 \pm 0.1791$ |
| U-Mamba Bot | $0.8380 \pm 0.0749$ | $0.8736 \pm 0.0646$ | $0.7977 \pm 0.1474$ |
| U-Mamba Enc | $0.8435 \pm 0.0725$ | $0.8537 \pm 0.0810$ | $0.7737 \pm 0.1543$ |
| nnU-Net | $0.8353 \pm 0.0927$ | $0.8633 \pm 0.0768$ | $0.7944 \pm 0.1474$ |
| **Domain generalization variants (nnU-Net backbone)** | | | |
| GroupDRO | $0.8565 \pm 0.0669$ | $0.8764 \pm 0.0602$ | $0.7929 \pm 0.1525$ |
| IBERM | $0.8508 \pm 0.0692$ | $0.8640 \pm 0.0605$ | $0.7898 \pm 0.1473$ |
| RandConv | $0.8571 \pm 0.0652$ | $0.8743 \pm 0.0637$ | $0.7984 \pm 0.1465$ |
| SD | $0.8557 \pm 0.0664$ | $0.8754 \pm 0.0625$ | $0.7953 \pm 0.1502$ |
| VREX | $0.8548 \pm 0.0666$ | $0.8770 \pm 0.0599$ | $0.7905 \pm 0.1527$ |
| **Foundation models** | | | |
| MedSAM2 | $0.5966 \pm 0.1710$ | $0.5726 \pm 0.1931$ | $0.4751 \pm 0.1881$ |
| SAM-Med3D | $0.1431 \pm 0.1239$ | $0.0799 \pm 0.0932$ | $0.0935 \pm 0.0952$ |
| SAM-Med3D turbo | $0.4920 \pm 0.1752$ | $0.1419 \pm 0.1582$ | $0.3800 \pm 0.1596$ |
| TotalSegmentator | $0.4744 \pm 0.1958$ | $0.1518 \pm 0.1264$ | $0.2916 \pm 0.1654$ |

Table 2: Cross-sequence benchmark (Train on T1W → Test on T2W). $\dagger$ indicates statistically significant improvement over the nnU-Net baseline ($p < 0.05$).

| Model | Dice | NSD | HD95 (mm) |
|---|---|---|---|
| **General supervised segmentation models** | | | |
| 3D U-Net | $0.0033 \pm 0.0178$ | $0.0017 \pm 0.0057$ | $115.69 \pm 52.39$ |
| SegResNet | $0.0129 \pm 0.0346$ | $0.0056 \pm 0.0101$ | $81.16 \pm 41.25$ |
| SwinUNETR | $0.0147 \pm 0.0383$ | $0.0070 \pm 0.0119$ | $93.43 \pm 39.63$ |
| SegMamba | $0.0014 \pm 0.0070$ | $0.0012 \pm 0.0046$ | $91.61 \pm 31.97$ |
| U-Mamba Bot | $0.0034 \pm 0.0146$ | $0.0142 \pm 0.0240$ | $112.66 \pm 45.29$ |
| U-Mamba Enc | $0.0096 \pm 0.0481$ | $0.0132 \pm 0.0289$ | $110.54 \pm 39.78$ |
| nnU-Net | $0.00027 \pm 0.00166$ | $0.0110 \pm 0.0302$ | $390.54 \pm 213.62$ |
| **Domain generalization variants (nnU-Net backbone)** | | | |
| GroupDRO | $0.00045 \pm 0.00211$ | $0.0169 \pm 0.0484$ | $374.93 \pm 184.35$ |
| IBERM$^\dagger$ | $0.00131 \pm 0.00767$ | $0.0209 \pm 0.0533$ | $382.19 \pm 242.41$ |
| RandConv | $0.00025 \pm 0.00128$ | $0.0141 \pm 0.0265$ | $446.62 \pm 256.62$ |
| SD$^\dagger$ | $0.00030 \pm 0.00149$ | $0.0141 \pm 0.0339$ | $437.86 \pm 258.09$ |
| VREX | $0.00039 \pm 0.00201$ | $0.0134 \pm 0.0321$ | $392.91 \pm 232.79$ |
| **Foundation models** | | | |
| MedSAM2 | $0.5726 \pm 0.1931$ | $0.6689 \pm 0.1744$ | $37.86 \pm 29.97$ |
| SAM-Med3D | $0.0798 \pm 0.0931$ | $0.0496 \pm 0.0684$ | $79.94 \pm 26.25$ |
| SAM-Med3D turbo | $0.1418 \pm 0.1582$ | $0.0799 \pm 0.0430$ | $50.52 \pm 26.84$ |
| TotalSegmentator | $0.1518 \pm 0.1264$ | $0.2862 \pm 0.1595$ | $220.47 \pm 110.80$ |

Notably, foundation models evaluated in zero-shot mode exhibit a remarkable resilience where specialized models fail. MedSAM2 (Zero-shot) achieves a Dice score of 0.5726 on the catastrophic T1W $\rightarrow$ T2W transfer, outperforming the collapsed supervised baseline (Dice $\approx 0.00$) by orders of magnitude. This reversal highlights a critical mechanism: supervised models fail because they overfit to sequence-specific intensity statistics, which are inverted in the target domain. In contrast, the frozen, large-scale pretrained encoders of foundation models rely on contrast-invariant shape priors and global objectness, effectively bypassing the texture-driven domain shift.

### 5.3. Multi-Sequence Joint Training

To understand whether the severe failures observed in Section 5.2 arise from distribution shifts rather than intrinsic data ambiguity, we evaluate an "oracle" setting in which models are jointly trained on all three sequences. As shown in Table 3, the nnU-Net baseline recovers from near-zero cross-sequence performance to a Dice of 0.818, demonstrating that the task itself is learnable once the contrast variability is exposed during training. This confirms that the collapse in single-source transfer is driven primarily by sequence-level distribution shifts.

Joint training further reveals architectural differences. U-Mamba Enc achieves the highest Dice (0.856), outperforming both convolutional and transformer-based models. This suggests that state-space architectures may better capture the multi-modal appearance distributions induced by heterogeneous MRI acquisition protocols. However, the gap across models narrows in this oracle setting, underscoring that data diversity is the dominant factor for achieving stable performance under sequence heterogeneity.

Table 3: Multi-sequence joint (All) benchmark results across all model families.

| Model | Dice | NSD | HD95 (mm) |
|---|---|---|---|
| General supervised segmentation models | | | |
| 3D U-Net | $0.7357 \pm 0.2037$ | $0.3439 \pm 0.1225$ | $15.83 \pm 25.87$ |
| SegResNet | $0.7856 \pm 0.1989$ | $0.4344 \pm 0.1379$ | $11.87 \pm 20.82$ |
| SwinUNETR | $0.7380 \pm 0.2065$ | $0.3676 \pm 0.1274$ | $16.51 \pm 23.75$ |
| SegMamba | $0.7976 \pm 0.1041$ | $0.5444 \pm 0.1486$ | $10.11 \pm 16.35$ |
| U-Mamba Bot | $0.8197 \pm 0.1107$ | $0.6575 \pm 0.1327$ | $26.77 \pm 48.84$ |
| U-Mamba Enc | $0.8557 \pm 0.0819$ | $0.7142 \pm 0.1184$ | $9.36 \pm 19.48$ |
| nnU-Net | $0.8178 \pm 0.0964$ | $0.7017 \pm 0.1525$ | $18.18 \pm 42.21$ |
| Domain generalization variants (nnU-Net backbone) | | | |
| GroupDRO | $0.8424 \pm 0.0872$ | $0.7403 \pm 0.1457$ | $12.54 \pm 40.08$ |
| IBERM | $0.8302 \pm 0.0927$ | $0.7266 \pm 0.1464$ | $17.50 \pm 55.09$ |
| RandConv | $0.8439 \pm 0.0890$ | $0.7470 \pm 0.1430$ | $11.15 \pm 37.52$ |
| SD | $0.8433 \pm 0.0859$ | $0.7444 \pm 0.1438$ | $12.11 \pm 39.92$ |
| VREX | $0.8391 \pm 0.0872$ | $0.7335 \pm 0.1479$ | $15.38 \pm 47.29$ |

### 5.4. Leave-One-Sequence-Out (LOSO)

We next examine whether increasing source diversity improves robustness. We report two representative settings: (i) training on T1W + OOP and testing on T2W (Table 4), and (ii) training on T2W + OOP and testing on T1W. Although the full results for the latter appear in Appendix F (Table 17), we analyze its behavior in the main text due to its representative value. The remaining configuration is included in Appendix F.

A clear pattern emerges. Compared with the single-source results in Section 5.2, simply adding the OOP sequence enables supervised models to recover from near-zero Dice (≈0.00–0.02) to substantially higher performance (e.g., nnU-Net: 0.1606, SegResNet: 0.3063). This demonstrates that exposing the model to any diversity in contrast mechanisms prevents the catastrophic overfitting to sequence-specific intensity rules observed in the single-source setting.

Interestingly, the LOSO setting also reveals that domain generalization algorithms become effective only when true multi-source diversity exists. For example, RandConv reaches 0.5537 in Table 17, outperforming the nnU-Net baseline (0.3258) by a large margin. This suggests that DG algorithms are ineffective when the domain shift is ill-defined (i.e., single-source training), but can meaningfully regularize the feature space once heterogeneous training sources are available.

Table 4: Leave-one-sequence-out benchmark (Train on T1W+OOP → Test on T2W).

| Model | Dice | NSD | HD95 (mm) |
|---|---|---|---|
| **General supervised segmentation models** | | | |
| 3D U-Net | $0.1586 \pm 0.1783$ | $0.0502 \pm 0.0573$ | $63.32 \pm 32.98$ |
| SegResNet | $0.3063 \pm 0.2353$ | $0.1060 \pm 0.0885$ | $55.57 \pm 39.75$ |
| SwinUNETR | $0.0762 \pm 0.0884$ | $0.0240 \pm 0.0212$ | $75.56 \pm 36.87$ |
| SegMamba | $0.1161 \pm 0.1630$ | $0.0506 \pm 0.0747$ | $64.03 \pm 39.65$ |
| U-Mamba Bot | $0.0919 \pm 0.1303$ | $0.0831 \pm 0.0830$ | $89.01 \pm 44.52$ |
| U-Mamba Enc | $0.1805 \pm 0.2143$ | $0.1129 \pm 0.1303$ | $86.17 \pm 48.11$ |
| nnU-Net | $0.1606 \pm 0.2043$ | $0.1819 \pm 0.1635$ | $239.24 \pm 191.06$ |
| **Domain generalization variants (nnU-Net backbone)** | | | |
| GroupDRO | $0.1669 \pm 0.1941$ | $0.2183 \pm 0.1624$ | $272.23 \pm 193.34$ |
| IBERM | $0.1169 \pm 0.1764$ | $0.1726 \pm 0.1752$ | $328.32 \pm 257.44$ |
| RandConv | $0.2673 \pm 0.2447$ | $0.2930 \pm 0.1877$ | $235.00 \pm 222.36$ |
| SD | $0.1957 \pm 0.2184$ | $0.2490 \pm 0.1916$ | $275.59 \pm 212.31$ |
| VREX | $0.1742 \pm 0.2043$ | $0.2249 \pm 0.1790$ | $238.12 \pm 204.23$ |
| **Foundation models** | | | |
| MedSAM2 | $0.5726 \pm 0.1931$ | $0.6689 \pm 0.1744$ | $37.86 \pm 29.97$ |
| SAM-Med3D | $0.0798 \pm 0.0931$ | $0.0496 \pm 0.0684$ | $79.94 \pm 26.25$ |
| SAM-Med3D turbo | $0.1418 \pm 0.1582$ | $0.0799 \pm 0.0430$ | $50.52 \pm 26.84$ |
| TotalSegmentator | $0.1518 \pm 0.1264$ | $0.2862 \pm 0.1595$ | $220.47 \pm 110.80$ |

### 5.5. Semi-Supervised Learning Performance

To evaluate whether semi-supervised learning can mitigate annotation scarcity in pancreas MRI segmentation, we examine two representative consistency-based SSL methods—UAMT and CPS—under four labeled ratios (5%, 10%, 20%, 40%). Experiments are conducted separately on T1W, T2W, and OOP sequences, and Figure 3 summarizes the Dice and HD95 trends.

SSL yields consistent improvements on the standard T1W sequence, confirming that consistency regularization is beneficial when intensity distributions are stable. However, the method exhibits pronounced instability on T2W imaging: HD95 for CPS spikes sharply at the 20% ratio, reflecting sensitivity to erroneous pseudo-labels. The effect is even more severe in the OOP sequence, where CPS collapses entirely at the 5% ratio (Dice < 0.1). Unlike T1W, the T2W pancreas exhibits higher intra-organ intensity variability, reduced boundary sharpness, and stronger overlap with surrounding fluid-tissue structures. These characteristics amplify model uncertainty and cause pseudo-labels to cluster around high-intensity but

anatomically ambiguous regions. In OOP imaging, fat-suppression artifacts and reduced dynamic range further degrade early pseudo-label reliability, making consistency-based optimization highly sensitive to noise. These results reveal that, unlike natural images, pancreas MRI requires a minimum level of reliable supervision; below this threshold, self-training reinforces structured errors and destabilizes optimization.

Overall, while SSL can reduce annotation demands under homogeneous protocols, it is not a guaranteed remedy for data-scarce, multi-contrast abdominal MRI. Careful monitoring and stronger pseudo-label filtering mechanisms are required before SSL can be reliably applied in this setting.

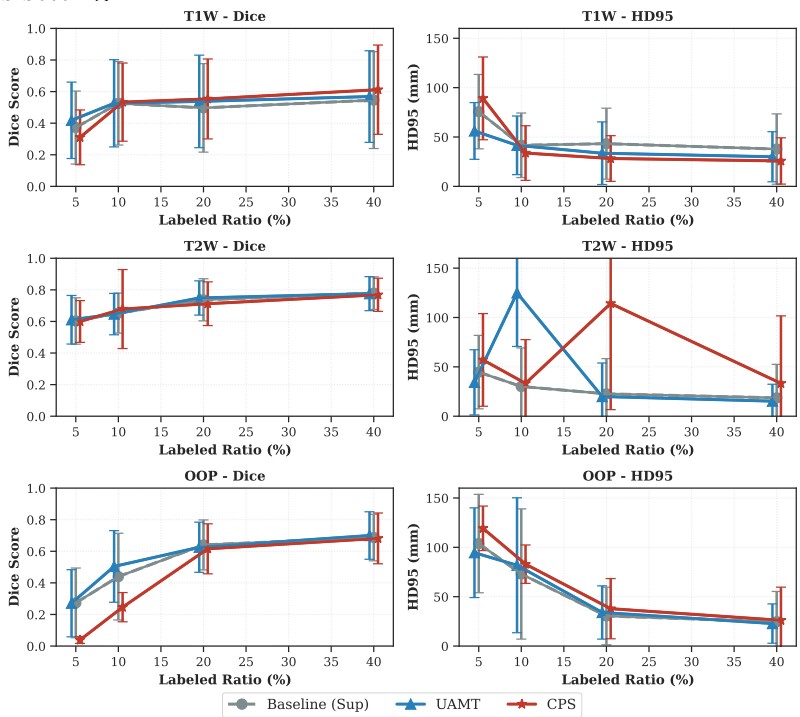

Figure 3: Performance of semi-supervised learning (SSL) methods across varying labeled ratios (5%, 10%, 20%, 40%) for T1W, T2W, and OOP sequences.

## 5.6. Ablation and Component Analysis

### 5.6.1. Cross-Center vs Cross-Sequence Behavior

To isolate whether the severe degradation observed in cross-sequence settings stems from sequence differences rather than center variations, we conduct a controlled cross-center ablation. We use T1W scans from MCF and NYU as the source domain and evaluate the trained models on other T1W scans and OOP scans. We do not include T2W in this analysis because earlier experiments (Section 5.2) showed that T1W → T2W performance collapses to near-zero, leaving little room for cross-center interpretation.

Figure 4 examines whether cross-center variability reflects a genuine distribution shift or simply data scarcity. We observe that increasing training data consistently reduces Dice discrepancies between centers and strengthens the correlation between validation and target

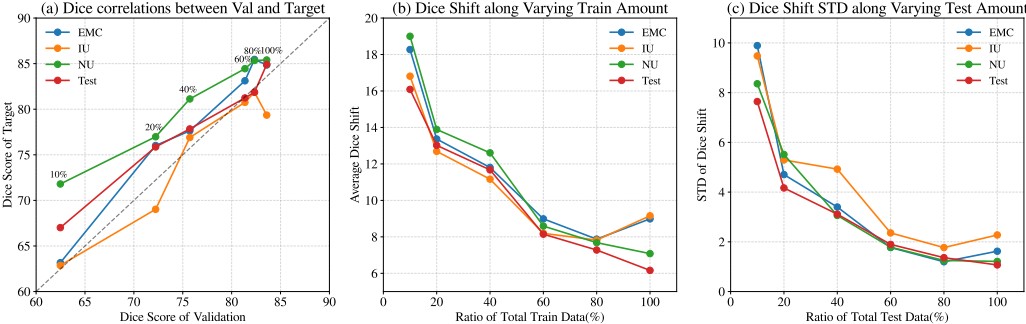

Figure 4: Cross-center behavior under varying amounts of training and test data. (a) Dice correlations between validation and target centers. (b) Average Dice shift decreases substantially as the training set grows. (c) Increasing the test-set size also reduces shift variance.

performance. Moreover, the variance of Dice shift across test samples decreases as test-set size grows, suggesting that part of the cross-center "domain gap" reported in prior studies may arise from small-sample effects at the case level rather than distribution differences.

In contrast, Figure 5 further highlights the distinction between center-level and sequence-level shifts. With prolonged training (100–1000 epochs), same-sequence (T1W → T1W) performance across centers continues to improve, consistent with classical overfitting on homogeneous contrast. In contrast, cross-phase (T1W → OOP) performance plateaus or even deteriorates, reaffirming that sequence shifts constitute a substantially more challenging generalization barrier that does not benefit from additional training or model capacity.

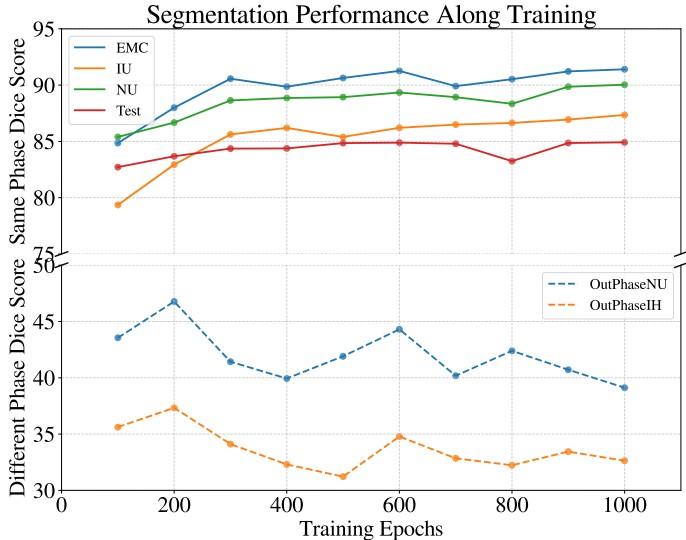

Figure 5: Comparing cross-center and cross-sequence behavior during extended training (100–1000 epochs).

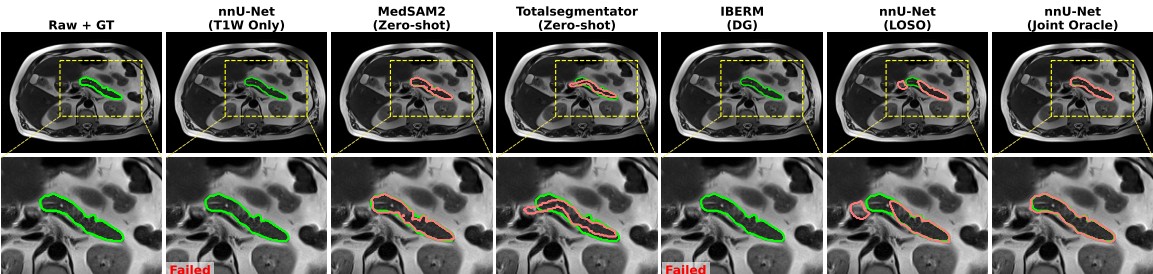

Figure 6: Qualitative comparison of pancreas MRI segmentation under the T1W → T2W sequence shift. Columns show (left → right): ground-truth annotation, nnU-Net trained only on T1W (single-source baseline), zero-shot MedSAM2, zero-shot Totalsegmentator, IBERM (DG method), nnU-Net trained via Leave-One-Sequence-Out (LOSO), and nnU-Net trained jointly on all sequences (Oracle). Green: Ground Truth; Red: Model Prediction.

### 5.6.2. QUALITATIVE ANALYSIS

To complement our quantitative evaluation, Figure 6 provides a qualitative comparison illustrating the characteristic failure modes that arise under the severe T1W → T2W sequence shift. More results can be found in Appendix G.

When trained only on T1W (Column 2), the baseline nnU-Net exhibits a complete functional collapse, producing no valid foreground mask. This confirms that supervised models overfit strongly to T1W-specific intensity relationships and cannot generalize once these relationships invert in T2-weighted imaging.

Similarly, the DG method IBERM (Column 5) fails to recover meaningful pancreas structure, indicating that statistical alignment techniques—which assume covariate-style shifts—are insufficient when the underlying image formation physics changes nonlinearly across sequences.

In contrast, MedSAM2 (Column 3) and Totalsegmentator (Column 4) successfully localize the pancreas despite the drastic contrast mismatch. Although the boundaries deviate from the ground truth, the predictions remain structurally coherent. This suggests that foundation models rely less on local intensity cues and more on global anatomical priors and shape representations learned from large-scale pretraining.

Finally, the LOSO model (Column 6), which incorporates OOP variation during training, restores consistent organ delineation and closely approaches the Oracle joint-training model (Column 6). This visual evidence supports our quantitative findings that exposure to any form of contrast diversity is key to mitigating sequence-induced domain shifts. Overall, these qualitative observations highlight a fundamental pattern: sequence shifts break low-level texture learners (CNNs, DG models), whereas contrast diversity or strong anatomical priors (LOSO, foundation models) are required to achieve stable generalization.

### 5.6.3. FOUNDATION MODELS ANALYSIS

Our ablation in Table 5 reveals three distinct transfer behaviors among foundational segmentation models.

**MedSAM2 exhibits strong and stable zero-shot performance.** Across all three MRI sequences, MedSAM2 achieves Dice scores of 0.47–0.59 without any fine-tuning. Inter-

Table 5: Ablation of foundation models. Dice scores (mean ± std) for different tuning strategies across T1W, T2W, and OOP sequences are reported.

| Model | T1W Dice | T2W Dice | OOP Dice |
|---|---|---|---|
| **MedSAM2** | | | |
| Zero-shot | $0.5966 \pm 0.1710$ | $0.5726 \pm 0.1931$ | $0.4751 \pm 0.1881$ |
| Full fine-tuning | $0.4861 \pm 0.1842$ | $0.4005 \pm 0.1573$ | $0.4759 \pm 0.0921$ |
| Decoder-only fine-tuning | $0.3977 \pm 0.2312$ | $0.4779 \pm 0.1626$ | $0.3787 \pm 0.1865$ |
| **SAM-Med3D** | | | |
| Zero-shot | $0.1431 \pm 0.1239$ | $0.0799 \pm 0.0932$ | $0.0935 \pm 0.0952$ |
| Full fine-tuning | $0.5282 \pm 0.1634$ | $0.4790 \pm 0.1737$ | $0.3769 \pm 0.1768$ |
| Decoder-only fine-tuning | $0.5236 \pm 0.1172$ | $0.5089 \pm 0.1205$ | $0.4256 \pm 0.0988$ |
| **SAM-Med3D-Turbo** | | | |
| Zero-shot | $0.4920 \pm 0.1752$ | $0.1419 \pm 0.1582$ | $0.3800 \pm 0.1596$ |
| Full fine-tuning | $0.5919 \pm 0.1554$ | $0.4937 \pm 0.1576$ | $0.5515 \pm 0.1634$ |
| Decoder-only fine-tuning | $0.5927 \pm 0.0956$ | $0.5587 \pm 0.1186$ | $0.5821 \pm 0.0762$ |
| **Totalsegmentator** | | | |
| Zero-shot | $0.4743 \pm 0.1957$ | $0.1518 \pm 0.1263$ | $0.2916 \pm 0.1653$ |
| Full fine-tuning | $0.8398 \pm 0.0731$ | $0.7707 \pm 0.1162$ | $0.7773 \pm 0.1497$ |
| Decoder-only fine-tuning | $0.8488 \pm 0.0677$ | $0.8229 \pm 0.0833$ | $0.7792 \pm 0.1485$ |

estingly, full fine-tuning degrades performance, and decoder-only tuning further decreases accuracy. This suggests that MedSAM2's large-scale image pretraining already provides a strong anatomical prior, and updating the parameters disrupts its learned invariances.

**SAM-Med3D collapses in zero-shot settings.** Unlike MedSAM2, SAM-Med3D achieves Dice $\approx 0.1$ in zero-shot mode, indicating that the model has poor transferability. However, both full fine-tuning and decoder-only tuning dramatically improve performance, showing that SAM-Med3D requires supervised adaptation to become functional.

**SAM-Med3D-Turbo behaves between the two extremes.** Turbo provides moderate zero-shot performance and benefits from decoder-only fine-tuning more than full fine-tuning. This pattern indicates that Turbo's hybrid pretraining partially transfers to MRI, but still needs mild adaptation.

**TotalSegmentator requires adaptation despite MRI pretraining.** The MRI-specific TotalSegmentator exhibits a distinct behavior from the SAM family. Despite being pretrained on large-scale MRI datasets, its zero-shot transferability to our pancreas cohort is limited. However, both full fine-tuning and decoder-only tuning trigger a massive performance recovery, effectively matching the best supervised baselines. This suggests that its frozen representations are not perfectly aligned with our heterogeneous scans. It functions best as a highly effective weight initialization, requiring supervised gradients to realign its pre-learned MRI features to the target distribution.

These findings highlight that SAM-based models are not interchangeable: MedSAM2 is inherently robust and transferable, whereas 3D SAM variants rely heavily on task-specific fine-tuning. In contrast, domain-specific pretraining like TotalSegmentator does not guarantee zero-shot transferability but provides a high-quality initialization that demands supervised adaptation to bridge the final modality gap. This distinction emphasizes the need to evaluate foundation models individually rather than assuming uniform behavior across the family.

To further explore the limits of foundation models, we provide additional comparisons with the state-of-the-art VISTA3D (He et al., 2025) and nnInteractive (Isensee et al., 2025) in Appendix H. Remarkably, both achieve near-supervised performance. These results indi-

cate that while physics-driven sequence shifts lead to severe failures in standard supervised models, such effects can be partially mitigated through large-scale pretraining.

### 5.7. External Validation

To reproduce the observations across sequences on external, publicly available datasets, we evaluated our trained models on the AMOS MRI dataset (Ji et al., 2022) with 60 cases. The results are shown in Table 6. The observed trends are highly consistent with our main findings. Models trained on CrossPan T1W generalize well to AMOS MRI, achieving Dice scores around 0.8. But models trained on T2W largely fail, and OOP-trained models exhibit intermediate performance. This pattern suggests that the major distribution of AMOS MRI is closer to T1W. It also proves that the sequence-dependent failure modes identified on CrossPan persist on external data.

We observe similar consistency among foundation models. MedSAM2, SAM-Med3D Turbo, and TotalSegmentator achieve around 0.5 Dice under zero-shot inference, while SAM-Med3D exhibits substantially weaker transfer. The results closely mirror their behavior on CrossPan. It further supports the conclusion that large-scale pretraining induces contrast-invariant anatomical priors that partially mitigate sequence-induced domain shifts. In all, these results indicate that the cross-sequence behaviors observed in our paper are not specific to CrossPan, but reflect a reproducible phenomenon on external clinical MRI data.

Table 6: External validation on the AMOS MRI dataset. Dice scores are presented.

| Model | Trained on T1W | Trained on T2W | Trained on OOP | Trained on all sequences |
|---|---|---|---|---|
| **General supervised segmentation models** | | | | |
| 3D U-Net | $0.7620 \pm 0.1930$ | $0.0218 \pm 0.0617$ | $0.1700 \pm 0.1940$ | $0.7743 \pm 0.1869$ |
| SegResNet | $0.7846 \pm 0.1975$ | $0.0043 \pm 0.0304$ | $0.2885 \pm 0.2437$ | $0.8179 \pm 0.1292$ |
| SwinUNETR | $0.7177 \pm 0.2471$ | $0.0204 \pm 0.0507$ | $0.2419 \pm 0.2107$ | $0.7990 \pm 0.1249$ |
| SegMamba | $0.8048 \pm 0.1582$ | $0.0040 \pm 0.0193$ | $0.3571 \pm 0.2730$ | $0.7759 \pm 0.1768$ |
| U-Mamba Bot | $0.7948 \pm 0.1325$ | $0.0127 \pm 0.0416$ | $0.5190 \pm 0.2384$ | $0.7947 \pm 0.1307$ |
| U-Mamba Enc | $0.8263 \pm 0.1219$ | $0.0044 \pm 0.0206$ | $0.4040 \pm 0.2624$ | $0.8163 \pm 0.1419$ |
| nnU-Net | $0.8164 \pm 0.1541$ | $0.0125 \pm 0.0663$ | $0.6253 \pm 0.2376$ | $0.8257 \pm 0.1213$ |
| **Domain generalization variants (nnU-Net backbone)** | | | | |
| GroupDRO | $0.8435 \pm 0.1180$ | $0.0216 \pm 0.0791$ | $0.5577 \pm 0.2551$ | $0.8298 \pm 0.1069$ |
| IBERM | $0.8179 \pm 0.1560$ | $0.0148 \pm 0.0720$ | $0.5583 \pm 0.2314$ | $0.8217 \pm 0.1269$ |
| RandConv | $0.8429 \pm 0.1279$ | $0.0338 \pm 0.1111$ | $0.6561 \pm 0.1995$ | $0.8363 \pm 0.1181$ |
| SD | $0.8385 \pm 0.1393$ | $0.0097 \pm 0.0531$ | $0.5839 \pm 0.2300$ | $0.8317 \pm 0.1187$ |
| VREX | $0.8399 \pm 0.1287$ | $0.0063 \pm 0.0368$ | $0.5671 \pm 0.2299$ | $0.8267 \pm 0.1215$ |
| **Foundation models** | | | | |
| MedSAM2 | | $0.5669 \pm 0.2066$ | | |
| SAM-Med3D | | $0.1356 \pm 0.1278$ | | |
| SAM-Med3D turbo | | $0.6075 \pm 0.1637$ | | |
| TotalSegmentator | | $0.4698 \pm 0.1921$ | | |

## 6. Discussion

Our study provides the first systematic evidence that sequence variability—rather than model capacity or center differences—is the primary obstacle for pancreas MRI segmentation. While state-of-the-art 3D architectures perform strongly in matched settings, their collapse under cross-sequence shifts reveals an over-reliance on sequence-specific intensity cues that do not transfer across contrast mechanisms. These findings highlight a fundamental gap: current models lack contrast-invariant representations essential for deployment in

heterogeneous clinical workflows. This need arises because MRI sequences differ fundamentally in their underlying signal formation physics, causing intensity patterns that are highly predictive in one sequence to become inverted or non-informative in another. Without representations that remain stable across these contrast mechanisms, models trained on one sequence inevitably overfit to sequence-specific appearance cues and fail to generalize.

A second insight is the limited effectiveness of existing DG and SSL methods for sequence shifts. DG methods improve robustness only when multi-sequence diversity is available, suggesting that statistical alignment alone cannot address physical contrast inversions. SSL exhibits similar limitations, with unstable behavior on T2W and OOP sequences. Together, these results indicate that robustness to MRI contrast variation requires domain-aware inductive biases, not simply larger data regimes or generic regularization.

While image-to-image translation, histogram matching, and other harmonization techniques can alleviate contrast-related domain shifts, these approaches typically rely on access to target-domain images or representative reference statistics. Such requirements limit their applicability in fully zero-shot or deployment-oriented scenarios. In our experiments, even lightweight harmonization such as histogram matching yielded negligible improvements, suggesting that intensity alignment alone is insufficient to bridge physics-driven sequence differences. For this reason, our study focuses on Domain Generalization and Foundation Models, and we consider explicit style-transfer–based harmonization as an important direction for future work.

Third, the performance of foundation models reveals a nuanced picture: MedSAM2 demonstrates strong zero-shot stability, whereas SAM-Med3D requires finetuning. The key difference lies in the pretraining data scale and composition. MedSAM2 is pretrained on both 2D and 3D data with 455,000 3D images and 76,000 frames. This breadth encourages the model to learn shape-centric representations. In contrast, SAM-Med3D is pretrained primarily on curated 3D datasets with only 22K 3D images, which are less diverse than the ones used for MedSAM2. As a result, SAM-Med3D relies more heavily on local appearance cues, leading to degraded zero-shot transfer. This divergence underscores the importance of understanding pretraining domain composition and representation biases, rather than assuming all foundation models generalize equally.

Finally, under this benchmark, a validated model is not defined only by high in-domain accuracy, but also by stable performance across different sequences. Specifically, a deployable model should achieve reasonable segmentation performance across sequences, rather than exhibiting near-complete failure under distribution shifts. From a deployment perspective, our findings suggest a practical guideline: segmentation models trained on a single MRI sequence should not be used in heterogeneous clinical settings unless they have been tested across multiple sequences using CrossPan. Without such validation, these models may fail when applied to scans acquired with a different protocol. In this way, CrossPan acts as a gatekeeper to detect and prevent these failure cases before deployment.

## 7. Conclusion

**CrossPan** provides the first systematic evaluation of cross-sequence generalization in pancreas MRI segmentation. Our experiments reveal that sequence-induced domain shifts—not model capacity or center diversity—constitute the primary barrier to robust performance:

models achieving Dice $> 0.85$ in-domain collapse to $< 0.02$ under cross-sequence transfer. Domain generalization methods designed for style-based shifts fail under physics-driven contrast inversions, while foundation models show divergent behavior depending on pretraining composition. Semi-supervised learning improves performance only under stable intensity distributions.

These findings have direct implications for clinical deployment. A pancreas segmentation system trained on a single MRI protocol cannot be safely applied to other sequences without validation or adaptation. We release **CrossPan** to support the development of contrast-robust learning strategies and to establish evaluation standards for heterogeneous abdominal MRI. Future work should explore physics-informed representation learning, sequence-conditioned architectures, and hybrid foundation-model adaptation as paths toward clinically reliable cross-sequence generalization.

## Acknowledgments

This study was funded by NIH R01-CA246704.

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

## Appendix A. Dataset Details

Here, we present the detailed composition of our dataset in Table 7.

Table 7: Statistics of sample distribution across different MRI sequences and centers.

| Sequence | Center | Samples |
|---|---|---|
| T1-weighted (T1W) | MCF | 151 |
| | NYU | 162 |
| | EMC | 50 |
| | IU | 50 |
| | NU | 50 |
| T2-weighted (T2W) | MCF | 143 |
| | NYU | 162 |
| | EMC | 102 |
| | IU | 73 |
| | NU | 207 |
| | AHN | 27 |
| | MCA | 23 |
| Out of Phase (OOP) | NU | 100 |
| | EMC | 36 |
| | IH | 50 |

## Appendix B. Evaluated Model Architectures

**Classical 3D architectures.** We include widely-used 3D segmentation models including 3D U-Net (Ronneberger et al., 2015), SegResNet (Myronenko, 2018), SwinUNETR (Hatamizadeh et al., 2021), SegMamba (Xing et al., 2024), and U-Mamba (Ma et al., 2024). These models represent standard choices for volumetric medical imaging and serve as strong supervised baselines. All these models are trained from scratch using official configurations within a unified nnU-Net framework.

**Domain generalization (DG) methods.** To assess robustness under sequence-induced distribution shifts, we evaluate representative DG algorithms: GroupDRO (Sagawa et al., 2019), spectral decoupling (SD) (Pezeshki et al., 2021), V-REx (Krueger et al., 2021), IB-ERM (Ahuja et al., 2021), and RandConv (Xu et al., 2021). Each method is applied on top of a 3D nnU-Net (Isensee et al., 2021) backbone following standard DG training protocols.

**SAM-based medical segmentation models.** We evaluate MedSAM2 (Ma et al., 2025) and SAM-Med3D (Wang et al., 2024) under direct inference (zero-shot) and fine-tuning. During fine-tuning, point prompts are automatically generated from ground-truth masks. These models aim to extend promptable image segmentation to medical imaging and provide a contrast-agnostic comparison point.

**Pretrained MRI segmentation models.** We include the TotalSegmentator MRI (Wasserthal et al., 2023) foundation model, evaluated under both zero-shot inference and task-specific finetuning, to assess the current state of pretrained volumetric models in cross-sequence pancreas MRI.

**Semi-supervised learning (SSL).** For the limited-label setting, we consider two state-of-the-art SSL strategies: Uncertainty-Aware Mean Teacher (UA-MT) (Yu et al., 2019) and Cross Pseudo Supervision (CPS) (Chen et al., 2021). Both methods leverage unlabeled data and consistency regularization to improve segmentation quality in limited-annotation settings.

In our comparisons, nnU-Net serves as a self-configuring clinical baseline; other CNN-, transformer-, and Mamba-based models represent diverse supervised architectures with varying inductive biases; domain generalization methods reflect state-of-the-art strategies for mitigating distribution shift; and foundation models offer a reference that is more robust to contrast variations.

Together, these method families collectively capture the landscape of modern medical segmentation approaches and allow us to compare their robustness under heterogeneous MRI sequences.

## Appendix C. Training Protocols

**Overview.** To ensure fair comparison, all task-specific models and DG variants were implemented within a unified nnU-Net framework, ensuring identical preprocessing, data augmentation, and inference pipelines. Each model was trained using a patch size of 48 $\times$ 192 $\times$ 192, SGD optimizer with Nesterov momentum ($\mu = 0.99$) and an initial learning rate of 0.01 for 400 epochs. The batch size was fixed to 2. No early stopping was employed to ensure convergence across all baselines. All experiments were conducted on a cluster equipped with NVIDIA A100 (80GB) GPUs. To ensure reproducibility, we fixed random seeds for data splitting and network initialization.

**Data splits.** To avoid information leakage across institutions and to keep center distributions balanced, we first perform splits within each center. For every center–sequence combination, cases are divided into 70% training, 10% validation, and 20% test at the patient level. After this center-wise split, we pool all centers that share the same MRI sequence (T1W, T2W, or OOP) to form the global train/validation/test sets used in our experiments. The resulting sequence-specific splits are fixed and reused across all evaluation settings and model families. Data splitting was strictly performed at the patient level to ensure that volumes from the same subject never appear across training, validation, and test sets, thereby preventing data leakage.

**Preprocessing and augmentation.** All volumes were resampled to a voxel spacing of $3 \times 1.25 \times 1.25$ mm. Online augmentations include random flips, random rotations, random crops, and intensity shifts. We applied only per-volume z-score normalization and did not use harmonization methods to avoid artificially reducing sequence or center variability.

**DG protocol.** Each DG method is trained according to its standard objective with no access to target-sequence data. For single-source settings, groups were defined based on centers. For multi-source settings, groups were defined based on sequences.

**SAM-based and pretrained models.** MedSAM2, SAM-Med3D, and TotalSegmentator MRI are evaluated in two modes: (1) zero-shot inference, using the pretrained checkpoint without further tuning; (2) finetuning, where models are trained with default prompts generated from ground truth (SAM-based) or ground truth masks (TotalSegmentator), consistent with their original designs.

**SSL protocol.** UA-MT and CPS are trained using labeled subsets containing 5%, 10%, 20%, or 40% of each sequence, with the remaining images treated as unlabeled. Unlabeled data contribute only through consistency losses. For UA-MT, the student network was updated via EMA with momentum 0.99, and consistency loss weight followed a ramp-up schedule over the first 50 epochs. For CPS, both networks were updated using the cross-pseudo label consistency with weight $\lambda(t)$ increasing linearly from 0 to 1.0 during the first 50 epochs.

## Appendix D. In-Domain Baseline

This section provides the complete Dice, HD95, and NSD metrics for all models under in-domain training, complementing the summary in Section 5.1. Table 8, Table 9, and Table 10 detail the performances for T1W, T2W, and OOP sequences, respectively.

Table 8: In-domain T1W benchmark results across all model families.

| Model | Dice | NSD | HD95 (mm) |
|---|---|---|---|
| **General supervised segmentation models** | | | |
| 3D U-Net | $0.6606 \pm 0.2901$ | $0.2932 \pm 0.1462$ | $19.04 \pm 22.92$ |
| SegResNet | $0.6870 \pm 0.2974$ | $0.3347 \pm 0.1615$ | $19.42 \pm 24.69$ |
| SwinUNETR | $0.6685 \pm 0.2921$ | $0.3266 \pm 0.1597$ | $20.48 \pm 23.98$ |
| SegMamba | $0.8325 \pm 0.0760$ | $0.5441 \pm 0.1225$ | $7.32 \pm 8.64$ |
| U-Mamba Bot | $0.8380 \pm 0.0749$ | $0.6486 \pm 0.1051$ | $14.21 \pm 29.42$ |
| U-Mamba Enc | $0.8435 \pm 0.0725$ | $0.6591 \pm 0.0995$ | $8.77 \pm 14.10$ |
| nnU-Net | $0.8353 \pm 0.0927$ | $0.6963 \pm 0.1266$ | $14.43 \pm 54.14$ |
| **Domain generalization variants (nnU-Net backbone)** | | | |
| GroupDRO | $0.8565 \pm 0.0669$ | $0.7191 \pm 0.1147$ | $14.46 \pm 59.94$ |
| IBERM | $0.8508 \pm 0.0692$ | $0.7085 \pm 0.1180$ | $14.69 \pm 59.26$ |
| RandConv | $0.8571 \pm 0.0652$ | $0.7200 \pm 0.1134$ | $13.68 \pm 54.35$ |
| SD | $0.8557 \pm 0.0664$ | $0.7155 \pm 0.1167$ | $15.11 \pm 58.13$ |
| VREX | $0.8548 \pm 0.0666$ | $0.7169 \pm 0.1155$ | $13.45 \pm 57.26$ |
| **Foundation models** | | | |
| **MedSAM2** | | | |
| Zero-shot | $0.5966 \pm 0.1710$ | $0.6749 \pm 0.1646$ | $34.32 \pm 23.31$ |
| Full fine-tuning | $0.4861 \pm 0.1842$ | $0.5714 \pm 0.1825$ | $47.75 \pm 23.42$ |
| Decoder-only fine-tuning | $0.3977 \pm 0.2312$ | $0.4972 \pm 0.2185$ | $58.65 \pm 33.26$ |
| **SAM-Med3D** | | | |
| Zero-shot | $0.1431 \pm 0.1240$ | $0.0696 \pm 0.0432$ | $75.95 \pm 29.14$ |
| Full fine-tuning | $0.5282 \pm 0.1634$ | $0.2156 \pm 0.0706$ | $22.34 \pm 13.51$ |
| Decoder-only fine-tuning | $0.5236 \pm 0.1172$ | $0.1778 \pm 0.0465$ | $14.10 \pm 6.56$ |
| **SAM-Med3D-Turbo** | | | |
| Zero-shot | $0.4920 \pm 0.1752$ | $0.1949 \pm 0.0757$ | $26.63 \pm 17.66$ |
| Full fine-tuning | $0.5919 \pm 0.1554$ | $0.2487 \pm 0.0720$ | $16.31 \pm 11.35$ |
| Decoder-only fine-tuning | $0.5927 \pm 0.0956$ | $0.2090 \pm 0.0485$ | $12.24 \pm 6.88$ |
| **TotalSegmentator** | | | |
| Zero-shot | $0.4744 \pm 0.1958$ | $0.3434 \pm 0.1308$ | $89.36 \pm 115.24$ |
| Full fine-tuning | $0.8398 \pm 0.0731$ | $0.6828 \pm 0.1280$ | $13.47 \pm 46.75$ |
| Decoder-only fine-tuning | $0.8489 \pm 0.0678$ | $0.7026 \pm 0.1202$ | $11.97 \pm 42.25$ |

Table 9: In-domain T2W benchmark results across all model families.

| Model | Dice | NSD | HD95 (mm) |
|---|---|---|---|
| **General supervised segmentation models** | | | |
| 3D U-Net | $0.8078 \pm 0.0997$ | $0.4275 \pm 0.1008$ | $13.96 \pm 28.87$ |
| SegResNet | $0.8421 \pm 0.0755$ | $0.5036 \pm 0.1041$ | $9.14 \pm 20.70$ |
| SwinUNETR | $0.8168 \pm 0.0998$ | $0.4779 \pm 0.1070$ | $11.93 \pm 21.59$ |
| SegMamba | $0.8298 \pm 0.0875$ | $0.6418 \pm 0.1377$ | $10.55 \pm 26.35$ |
| U-Mamba Bot | $0.8736 \pm 0.0646$ | $0.7690 \pm 0.1013$ | $12.79 \pm 35.01$ |
| U-Mamba Enc | $0.8537 \pm 0.0810$ | $0.7362 \pm 0.1117$ | $16.49 \pm 39.79$ |
| nnU-Net | $0.8633 \pm 0.0768$ | $0.8050 \pm 0.1184$ | $7.46 \pm 19.71$ |
| **Domain generalization variants (nnU-Net backbone)** | | | |
| GroupDRO | $0.8764 \pm 0.0602$ | $0.8224 \pm 0.1129$ | $11.73 \pm 49.35$ |
| IBERM | $0.8640 \pm 0.0605$ | $0.8000 \pm 0.1164$ | $16.88 \pm 56.82$ |
| RandConv | $0.8743 \pm 0.0637$ | $0.8190 \pm 0.1140$ | $11.26 \pm 46.77$ |
| SD | $0.8754 \pm 0.0625$ | $0.8216 \pm 0.1123$ | $12.32 \pm 48.09$ |
| VREX | $0.8770 \pm 0.0599$ | $0.8229 \pm 0.1128$ | $10.36 \pm 46.70$ |
| **Foundation models** | | | |
| **MedSAM2** | | | |
|     Zero-shot | $0.5726 \pm 0.1931$ | $0.6689 \pm 0.1744$ | $37.86 \pm 29.97$ |
|     Full fine-tuning | $0.4005 \pm 0.1573$ | $0.4843 \pm 0.1532$ | $61.11 \pm 32.83$ |
|     Decoder-only fine-tuning | $0.4779 \pm 0.1626$ | $0.5815 \pm 0.1667$ | $50.72 \pm 28.98$ |
| **SAM-Med3D** | | | |
|     Zero-shot | $0.0799 \pm 0.0932$ | $0.0497 \pm 0.0344$ | $79.95 \pm 26.26$ |
|     Full fine-tuning | $0.4790 \pm 0.1737$ | $0.1918 \pm 0.0684$ | $20.63 \pm 12.65$ |
|     Decoder-only fine-tuning | $0.5089 \pm 0.1205$ | $0.1671 \pm 0.0445$ | $18.33 \pm 12.09$ |
| **SAM-Med3D-Turbo** | | | |
|     Zero-shot | $0.1419 \pm 0.1582$ | $0.0800 \pm 0.0431$ | $50.52 \pm 26.85$ |
|     Full fine-tuning | $0.4937 \pm 0.1576$ | $0.1941 \pm 0.0632$ | $18.64 \pm 11.43$ |
|     Decoder-only fine-tuning | $0.5587 \pm 0.1186$ | $0.1948 \pm 0.0461$ | $12.96 \pm 10.49$ |
| **TotalSegmentator** | | | |
|     Zero-shot | $0.1518 \pm 0.1264$ | $0.2862 \pm 0.1595$ | $220.47 \pm 110.80$ |
|     Full fine-tuning | $0.7707 \pm 0.1162$ | $0.6943 \pm 0.1512$ | $34.09 \pm 95.66$ |
|     Decoder-only fine-tuning | $0.8229 \pm 0.0834$ | $0.7496 \pm 0.1358$ | $14.73 \pm 35.63$ |

Table 10: In-domain OOP benchmark results across all model families.

| Model | Dice | NSD | HD95 (mm) |
|---|---|---|---|
| **General supervised segmentation models** | | | |
| 3D U-Net | $0.6729 \pm 0.1596$ | $0.2369 \pm 0.0748$ | $16.52 \pm 9.71$ |
| SegResNet | $0.7338 \pm 0.1517$ | $0.2957 \pm 0.0815$ | $13.82 \pm 10.87$ |
| SwinUNETR | $0.7040 \pm 0.1645$ | $0.3047 \pm 0.0945$ | $15.34 \pm 12.25$ |
| SegMamba | $0.7305 \pm 0.1791$ | $0.4772 \pm 0.1803$ | $12.01 \pm 9.96$ |
| U-Mamba Bot | $0.7977 \pm 0.1474$ | $0.6313 \pm 0.1456$ | $16.71 \pm 26.34$ |
| U-Mamba Enc | $0.7737 \pm 0.1543$ | $0.5982 \pm 0.1440$ | $17.32 \pm 23.41$ |
| nnU-Net | $0.7944 \pm 0.1474$ | $0.7131 \pm 0.1904$ | $20.27 \pm 46.66$ |
| **Domain generalization variants (nnU-Net backbone)** | | | |
| GroupDRO | $0.7929 \pm 0.1525$ | $0.7132 \pm 0.1957$ | $20.66 \pm 45.93$ |
| IBERM | $0.7898 \pm 0.1473$ | $0.7038 \pm 0.1923$ | $20.05 \pm 42.94$ |
| RandConv | $0.7984 \pm 0.1465$ | $0.7159 \pm 0.1888$ | $13.83 \pm 29.40$ |
| SD | $0.7953 \pm 0.1502$ | $0.7163 \pm 0.1896$ | $20.40 \pm 41.69$ |
| VREX | $0.7905 \pm 0.1527$ | $0.7065 \pm 0.1968$ | $17.71 \pm 37.77$ |
| **Foundation models** | | | |
| **MedSAM2** | | | |
| Zero-shot | $0.4751 \pm 0.1881$ | $0.5846 \pm 0.1780$ | $45.15 \pm 22.86$ |
| Full fine-tuning | $0.4759 \pm 0.0921$ | $0.6330 \pm 0.0955$ | $48.53 \pm 15.74$ |
| Decoder-only fine-tuning | $0.3787 \pm 0.1865$ | $0.4392 \pm 0.2056$ | $58.72 \pm 28.33$ |
| **SAM-Med3D** | | | |
| Zero-shot | $0.0935 \pm 0.0952$ | $0.0550 \pm 0.0337$ | $75.50 \pm 28.67$ |
| Full fine-tuning | $0.3769 \pm 0.1768$ | $0.1494 \pm 0.0506$ | $28.39 \pm 11.71$ |
| Decoder-only fine-tuning | $0.4256 \pm 0.0988$ | $0.1333 \pm 0.0314$ | $29.49 \pm 7.27$ |
| **SAM-Med3D-Turbo** | | | |
| Zero-shot | $0.3800 \pm 0.1596$ | $0.1332 \pm 0.0572$ | $35.59 \pm 19.54$ |
| Full fine-tuning | $0.5515 \pm 0.1634$ | $0.2070 \pm 0.0606$ | $15.59 \pm 7.86$ |
| Decoder-only fine-tuning | $0.5821 \pm 0.0762$ | $0.1883 \pm 0.0359$ | $16.68 \pm 9.06$ |
| **TotalSegmentator** | | | |
| Zero-shot | $0.2916 \pm 0.1654$ | $0.3187 \pm 0.1832$ | $165.54 \pm 127.22$ |
| Full fine-tuning | $0.7774 \pm 0.1498$ | $0.6903 \pm 0.1925$ | $21.53 \pm 49.32$ |
| Decoder-only fine-tuning | $0.7792 \pm 0.1486$ | $0.6902 \pm 0.1856$ | $24.90 \pm 52.97$ |

## Appendix E. Single-Source Cross-Sequence Generalization

To maintain clarity, the main text highlighted the most challenging generalization from T1W $\rightarrow$ T2W. Here, Table 11, Table 12, Table 13, Table 14, Table 15, and Table 16 present full results for all source $\rightarrow$ target combinations, allowing analysis of modality asymmetry, architecture sensitivity, and cross-sequence consistency.

Table 11: Cross-sequence benchmark (Train on T1W $\rightarrow$ Test on T2W). $^\dagger$ indicates statistically significant improvement over the nnU-Net baseline ($p < 0.05$).

| Model | Dice | NSD | HD95 (mm) |
|---|---|---|---|
| **General supervised segmentation models** | | | |
| 3D U-Net | $0.0033 \pm 0.0178$ | $0.0017 \pm 0.0057$ | $115.69 \pm 52.39$ |
| SegResNet | $0.0129 \pm 0.0346$ | $0.0056 \pm 0.0101$ | $81.16 \pm 41.25$ |
| SwinUNETR | $0.0147 \pm 0.0383$ | $0.0070 \pm 0.0119$ | $93.43 \pm 39.63$ |
| SegMamba | $0.0014 \pm 0.0070$ | $0.0012 \pm 0.0046$ | $91.61 \pm 31.97$ |
| U-Mamba Bot | $0.0034 \pm 0.0146$ | $0.0142 \pm 0.0240$ | $112.66 \pm 45.29$ |
| U-Mamba Enc | $0.0096 \pm 0.0481$ | $0.0132 \pm 0.0289$ | $110.54 \pm 39.78$ |
| nnU-Net | $0.00027 \pm 0.00166$ | $0.0110 \pm 0.0302$ | $390.54 \pm 213.62$ |
| **Domain generalization variants (nnU-Net backbone)** | | | |
| GroupDRO | $0.00045 \pm 0.00211$ | $0.0169 \pm 0.0484$ | $374.93 \pm 184.35$ |
| IBERM$^\dagger$ | $0.00131 \pm 0.00767$ | $0.0209 \pm 0.0533$ | $382.19 \pm 242.41$ |
| RandConv | $0.00025 \pm 0.00128$ | $0.0141 \pm 0.0265$ | $446.62 \pm 256.62$ |
| SD$^\dagger$ | $0.00030 \pm 0.00149$ | $0.0141 \pm 0.0339$ | $437.86 \pm 258.09$ |
| VREX | $0.00039 \pm 0.00201$ | $0.0134 \pm 0.0321$ | $392.91 \pm 232.79$ |
| **Foundation models** | | | |
| **MedSAM2** | | | |
| Zero-shot | $0.5726 \pm 0.1931$ | $0.6689 \pm 0.1744$ | $37.86 \pm 29.97$ |
| Full fine-tuning | $0.3036 \pm 0.1304$ | $0.4307 \pm 0.1153$ | $54.08 \pm 25.61$ |
| Decoder-only fine-tuning | $0.3420 \pm 0.1920$ | $0.4919 \pm 0.1672$ | $64.90 \pm 31.99$ |
| **SAM-Med3D** | | | |
| Zero-shot | $0.0799 \pm 0.0932$ | $0.0497 \pm 0.0344$ | $79.95 \pm 26.26$ |
| Full fine-tuning | $0.0672 \pm 0.0924$ | $0.0585 \pm 0.0325$ | $54.32 \pm 22.59$ |
| Decoder-only fine-tuning | $0.3418 \pm 0.1310$ | $0.1240 \pm 0.0375$ | $28.51 \pm 14.42$ |
| **SAM-Med3D Turbo** | | | |
| Zero-shot | $0.1419 \pm 0.1582$ | $0.0800 \pm 0.0431$ | $50.52 \pm 26.85$ |
| Full fine-tuning | $0.1108 \pm 0.1103$ | $0.0847 \pm 0.0344$ | $40.91 \pm 19.29$ |
| Decoder-only fine-tuning | $0.4333 \pm 0.1417$ | $0.1493 \pm 0.0439$ | $23.01 \pm 14.47$ |
| **TotalSegmentator** | | | |
| Zero-shot | $0.1518 \pm 0.1264$ | $0.2862 \pm 0.1595$ | $220.47 \pm 110.80$ |
| Full fine-tuning | $0.0018 \pm 0.0098$ | $0.0233 \pm 0.0481$ | $347.96 \pm 228.38$ |
| Decoder-only fine-tuning | $0.1180 \pm 0.1452$ | $0.1499 \pm 0.1212$ | $300.91 \pm 197.55$ |

Table 12: Cross-sequence benchmark (Train on T1W → Test on OOP). $^{\dagger}$ indicates statistically significant improvement over the nnU-Net baseline ($p < 0.05$).

| Model | Dice | NSD | HD95 (mm) |
|---|---|---|---|
| **General supervised segmentation models** | | | |
| 3D U-Net | $0.2956 \pm 0.2568$ | $0.0852 \pm 0.0868$ | $52.47 \pm 37.11$ |
| SegResNet | $0.2700 \pm 0.3023$ | $0.0998 \pm 0.1150$ | $52.34 \pm 29.96$ |
| SwinUNETR | $0.2938 \pm 0.2516$ | $0.0887 \pm 0.0912$ | $59.75 \pm 30.81$ |
| SegMamba | $0.2599 \pm 0.2579$ | $0.1290 \pm 0.1701$ | $47.35 \pm 26.99$ |
| U-Mamba Bot | $0.2986 \pm 0.2818$ | $0.1883 \pm 0.1990$ | $90.83 \pm 62.57$ |
| U-Mamba Enc | $0.3606 \pm 0.3075$ | $0.2276 \pm 0.2085$ | $64.88 \pm 43.14$ |
| nnU-Net | $0.2922 \pm 0.3064$ | $0.2788 \pm 0.2772$ | $175.20 \pm 158.26$ |
| **Domain generalization variants (nnU-Net backbone)** | | | |
| GroupDRO$^{\dagger}$ | $0.3239 \pm 0.3082$ | $0.3220 \pm 0.2738$ | $227.96 \pm 326.70$ |
| IBERM$^{\dagger}$ | $0.3367 \pm 0.3015$ | $0.2964 \pm 0.2746$ | $161.46 \pm 206.20$ |
| RandConv | $0.3229 \pm 0.2950$ | $0.3057 \pm 0.2667$ | $156.47 \pm 151.35$ |
| SD | $0.3337 \pm 0.3010$ | $0.3137 \pm 0.2878$ | $159.97 \pm 168.26$ |
| VREX$^{\dagger}$ | $0.3506 \pm 0.3017$ | $0.3291 \pm 0.2693$ | $198.55 \pm 256.22$ |
| **Foundation models** | | | |
| **MedSAM2** | | | |
|    Zero-shot | $0.4751 \pm 0.1881$ | $0.5846 \pm 0.1780$ | $45.15 \pm 22.86$ |
|    Full fine-tuning | $0.4318 \pm 0.1503$ | $0.5284 \pm 0.1710$ | $43.10 \pm 17.75$ |
|    Decoder-only fine-tuning | $0.3516 \pm 0.1885$ | $0.4761 \pm 0.1680$ | $60.30 \pm 31.57$ |
| **SAM-Med3D** | | | |
|    Zero-shot | $0.0935 \pm 0.0952$ | $0.0550 \pm 0.0337$ | $75.50 \pm 28.67$ |
|    Full fine-tuning | $0.3322 \pm 0.1990$ | $0.1266 \pm 0.0626$ | $32.83 \pm 16.30$ |
|    Decoder-only fine-tuning | $0.4532 \pm 0.1276$ | $0.1556 \pm 0.0421$ | $20.84 \pm 7.82$ |
| **SAM-Med3D Turbo** | | | |
|    Zero-shot | $0.3800 \pm 0.1596$ | $0.1332 \pm 0.0572$ | $35.59 \pm 19.54$ |
|    Full fine-tuning | $0.4487 \pm 0.1901$ | $0.1739 \pm 0.0674$ | $25.33 \pm 18.96$ |
|    Decoder-only fine-tuning | $0.5583 \pm 0.0887$ | $0.1898 \pm 0.0435$ | $16.45 \pm 8.23$ |
| **TotalSegmentator** | | | |
|    Zero-shot | $0.2916 \pm 0.1654$ | $0.3187 \pm 0.1832$ | $165.54 \pm 127.22$ |
|    Full fine-tuning | $0.4045 \pm 0.2996$ | $0.3490 \pm 0.2729$ | $135.38 \pm 156.04$ |
|    Decoder-only fine-tuning | $0.4856 \pm 0.2333$ | $0.3892 \pm 0.2334$ | $88.84 \pm 103.50$ |

Table 13: Cross-sequence benchmark (Train on T2W → Test on T1W). $^{\dagger}$ indicates statistically significant improvement over the nnU-Net baseline ($p < 0.05$).

| Model | Dice | NSD | HD95 (mm) |
|---|---|---|---|
| **General supervised segmentation models** | | | |
| 3D U-Net | $0.0084 \pm 0.0314$ | $0.0042 \pm 0.0141$ | $98.97 \pm 34.57$ |
| SegResNet | $0.0000 \pm 0.0001$ | $0.0000 \pm 0.0003$ | $147.77 \pm 54.64$ |
| SwinUNETR | $0.0121 \pm 0.0419$ | $0.0060 \pm 0.0154$ | $94.47 \pm 27.12$ |
| SegMamba | $0.0007 \pm 0.0041$ | $0.0004 \pm 0.0020$ | $103.62 \pm 30.45$ |
| U-Mamba Bot | $0.0117 \pm 0.0415$ | $0.0129 \pm 0.0261$ | $98.59 \pm 31.87$ |
| U-Mamba Enc | $0.0094 \pm 0.0361$ | $0.0109 \pm 0.0261$ | $86.08 \pm 28.18$ |
| nnU-Net | $0.0094 \pm 0.0276$ | $0.0290 \pm 0.0497$ | $295.17 \pm 213.47$ |
| **Domain generalization variants (nnU-Net backbone)** | | | |
| GroupDRO | $0.0066 \pm 0.0200$ | $0.0224 \pm 0.0507$ | $351.84 \pm 244.48$ |
| IBERM$^{\dagger}$ | $0.0212 \pm 0.0599$ | $0.0311 \pm 0.0580$ | $290.46 \pm 218.72$ |
| RandConv$^{\dagger}$ | $0.0228 \pm 0.0695$ | $0.0382 \pm 0.0682$ | $276.89 \pm 221.22$ |
| SD | $0.0149 \pm 0.0495$ | $0.0273 \pm 0.0526$ | $279.57 \pm 207.91$ |
| VREX | $0.0153 \pm 0.0574$ | $0.0263 \pm 0.0605$ | $293.04 \pm 215.27$ |
| **Foundation models** | | | |
| **MedSAM2** | | | |
| Zero-shot | $0.5966 \pm 0.1710$ | $0.6749 \pm 0.1646$ | $34.32 \pm 23.31$ |
| Full fine-tuning | $0.3222 \pm 0.1333$ | $0.3895 \pm 0.1376$ | $61.51 \pm 23.45$ |
| Decoder-only fine-tuning | $0.4329 \pm 0.1516$ | $0.5130 \pm 0.1568$ | $47.25 \pm 20.10$ |
| **SAM-Med3D** | | | |
| Zero-shot | $0.1431 \pm 0.1240$ | $0.0696 \pm 0.0432$ | $75.95 \pm 29.14$ |
| Full fine-tuning | $0.1131 \pm 0.1023$ | $0.0757 \pm 0.0335$ | $38.64 \pm 16.63$ |
| Decoder-only fine-tuning | $0.5294 \pm 0.1074$ | $0.1769 \pm 0.0499$ | $17.27 \pm 9.90$ |
| **SAM-Med3D Turbo** | | | |
| Zero-shot | $0.4920 \pm 0.1752$ | $0.1949 \pm 0.0757$ | $26.63 \pm 17.66$ |
| Full fine-tuning | $0.2634 \pm 0.1384$ | $0.1231 \pm 0.0408$ | $26.85 \pm 11.20$ |
| Decoder-only fine-tuning | $0.5581 \pm 0.1045$ | $0.1944 \pm 0.0460$ | $13.43 \pm 7.20$ |
| **TotalSegmentator** | | | |
| Zero-shot | $0.4744 \pm 0.1958$ | $0.3434 \pm 0.1308$ | $89.36 \pm 115.24$ |
| Full fine-tuning | $0.00002 \pm 0.00013$ | $0.0076 \pm 0.0177$ | $315.43 \pm 187.26$ |
| Decoder-only fine-tuning | $0.0962 \pm 0.1403$ | $0.1288 \pm 0.1019$ | $176.66 \pm 148.69$ |

Table 14: Cross-sequence benchmark (Train on T2W → Test on OOP). No statistically significant improvement over the nnU-Net baseline ($p < 0.05$).

| Model | Dice | NSD | HD95 (mm) |
|---|---|---|---|
| **General supervised segmentation models** | | | |
| 3D U-Net | $0.0706 \pm 0.1419$ | $0.0306 \pm 0.0537$ | $71.20 \pm 42.31$ |
| SegResNet | $0.0446 \pm 0.1104$ | $0.0158 \pm 0.0371$ | $131.05 \pm 62.16$ |
| SwinUNETR | $0.0434 \pm 0.0970$ | $0.0200 \pm 0.0393$ | $87.14 \pm 34.95$ |
| SegMamba | $0.0845 \pm 0.1426$ | $0.0571 \pm 0.0800$ | $72.46 \pm 32.56$ |
| U-Mamba Bot | $0.1030 \pm 0.2046$ | $0.0702 \pm 0.1289$ | $86.94 \pm 42.95$ |
| U-Mamba Enc | $0.1522 \pm 0.2342$ | $0.0929 \pm 0.1376$ | $70.69 \pm 46.06$ |
| nnU-Net | $0.1780 \pm 0.2510$ | $0.1813 \pm 0.2099$ | $277.07 \pm 297.96$ |
| **Domain generalization variants (nnU-Net backbone)** | | | |
| GroupDRO | $0.1497 \pm 0.2465$ | $0.1727 \pm 0.2060$ | $304.41 \pm 298.50$ |
| IBERM | $0.1332 \pm 0.2078$ | $0.1701 \pm 0.1714$ | $150.47 \pm 116.92$ |
| RandConv | $0.1657 \pm 0.2413$ | $0.1915 \pm 0.1940$ | $195.60 \pm 169.35$ |
| SD | $0.1422 \pm 0.2282$ | $0.1709 \pm 0.2005$ | $303.10 \pm 368.09$ |
| VREX | $0.1541 \pm 0.2407$ | $0.1714 \pm 0.2066$ | $304.69 \pm 361.85$ |
| **Foundation models** | | | |
| **MedSAM2** | | | |
|    Zero-shot | $0.4751 \pm 0.1881$ | $0.5846 \pm 0.1780$ | $45.15 \pm 22.86$ |
|    Full fine-tuning | $0.3384 \pm 0.1347$ | $0.4087 \pm 0.1110$ | $60.95 \pm 24.63$ |
|    Decoder-only fine-tuning | $0.4598 \pm 0.1094$ | $0.5644 \pm 0.1358$ | $44.79 \pm 20.93$ |
| **SAM-Med3D** | | | |
|    Zero-shot | $0.0935 \pm 0.0952$ | $0.0550 \pm 0.0337$ | $75.50 \pm 28.67$ |
|    Full fine-tuning | $0.2340 \pm 0.1413$ | $0.1098 \pm 0.0378$ | $29.13 \pm 12.56$ |
|    Decoder-only fine-tuning | $0.5312 \pm 0.1030$ | $0.1742 \pm 0.0457$ | $16.50 \pm 7.21$ |
| **SAM-Med3D Turbo** | | | |
|    Zero-shot | $0.3800 \pm 0.1596$ | $0.1332 \pm 0.0572$ | $35.59 \pm 19.54$ |
|    Full fine-tuning | $0.3105 \pm 0.1336$ | $0.1308 \pm 0.0299$ | $22.95 \pm 8.09$ |
|    Decoder-only fine-tuning | $0.5939 \pm 0.1157$ | $0.2065 \pm 0.0514$ | $12.37 \pm 6.86$ |
| **TotalSegmentator** | | | |
|    Zero-shot | $0.2916 \pm 0.1654$ | $0.3187 \pm 0.1832$ | $165.54 \pm 127.22$ |
|    Full fine-tuning | $0.0765 \pm 0.1135$ | $0.1820 \pm 0.1550$ | $277.37 \pm 260.89$ |
|    Decoder-only fine-tuning | $0.3458 \pm 0.2583$ | $0.3451 \pm 0.1977$ | $150.63 \pm 200.58$ |

Table 15: Cross-sequence benchmark (Train on OOP → Test on T1W). $^{\dagger}$ indicates statistically significant improvement over the nnU-Net baseline ($p < 0.05$).

| Model | Dice | NSD | HD95 (mm) |
|---|---|---|---|
| **General supervised segmentation models** | | | |
| 3D U-Net | $0.1342 \pm 0.1710$ | $0.0413 \pm 0.0523$ | $76.28 \pm 38.52$ |
| SegResNet | $0.1830 \pm 0.1935$ | $0.0555 \pm 0.0595$ | $57.39 \pm 23.39$ |
| SwinUNETR | $0.1630 \pm 0.2017$ | $0.0553 \pm 0.0670$ | $64.42 \pm 30.83$ |
| SegMamba | $0.2198 \pm 0.2303$ | $0.1000 \pm 0.1038$ | $62.44 \pm 30.15$ |
| U-Mamba Bot | $0.5103 \pm 0.2109$ | $0.2835 \pm 0.1327$ | $50.26 \pm 34.61$ |
| U-Mamba Enc | $0.3635 \pm 0.2581$ | $0.1846 \pm 0.1220$ | $63.07 \pm 37.52$ |
| nnU-Net | $0.5306 \pm 0.2394$ | $0.3773 \pm 0.1574$ | $113.06 \pm 132.51$ |
| **Domain generalization variants (nnU-Net backbone)** | | | |
| GroupDRO$^{\dagger}$ | $0.5357 \pm 0.2248$ | $0.3825 \pm 0.1501$ | $83.48 \pm 118.20$ |
| IBERM | $0.4901 \pm 0.2167$ | $0.3415 \pm 0.1273$ | $111.42 \pm 130.50$ |
| RandConv$^{\dagger}$ | $0.6095 \pm 0.1916$ | $0.4170 \pm 0.1406$ | $91.41 \pm 122.85$ |
| SD$^{\dagger}$ | $0.5385 \pm 0.2185$ | $0.3750 \pm 0.1495$ | $114.37 \pm 132.50$ |
| VREX | $0.5045 \pm 0.2279$ | $0.3781 \pm 0.1500$ | $104.57 \pm 128.01$ |
| **Foundation models** | | | |
| **MedSAM2** | | | |
|    Zero-shot | $0.5966 \pm 0.1710$ | $0.6749 \pm 0.1646$ | $34.32 \pm 23.31$ |
|    Full fine-tuning | $0.4445 \pm 0.1253$ | $0.5766 \pm 0.1195$ | $52.33 \pm 18.93$ |
|    Decoder-only fine-tuning | $0.3948 \pm 0.1926$ | $0.4614 \pm 0.2059$ | $60.70 \pm 28.00$ |
| **SAM-Med3D** | | | |
|    Zero-shot | $0.1431 \pm 0.1240$ | $0.0696 \pm 0.0432$ | $75.95 \pm 29.14$ |
|    Full fine-tuning | $0.2749 \pm 0.2070$ | $0.1114 \pm 0.0625$ | $36.30 \pm 19.87$ |
|    Decoder-only fine-tuning | $0.4431 \pm 0.1214$ | $0.1329 \pm 0.0361$ | $24.37 \pm 11.18$ |
| **SAM-Med3D Turbo** | | | |
|    Zero-shot | $0.4920 \pm 0.1752$ | $0.1949 \pm 0.0757$ | $26.63 \pm 17.66$ |
|    Full fine-tuning | $0.5272 \pm 0.1532$ | $0.1921 \pm 0.0614$ | $17.69 \pm 10.20$ |
|    Decoder-only fine-tuning | $0.5726 \pm 0.1112$ | $0.1967 \pm 0.0517$ | $13.72 \pm 8.07$ |
| **TotalSegmentator** | | | |
|    Zero-shot | $0.4744 \pm 0.1958$ | $0.3434 \pm 0.1308$ | $89.36 \pm 115.24$ |
|    Full fine-tuning | $0.5032 \pm 0.2563$ | $0.3586 \pm 0.1598$ | $101.14 \pm 156.46$ |
|    Decoder-only fine-tuning | $0.5359 \pm 0.2297$ | $0.4181 \pm 0.1499$ | $96.98 \pm 114.11$ |

Table 16: Cross-sequence benchmark (Train on OOP → Test on T2W). † indicates statistically significant improvement over the nnU-Net baseline ($p < 0.05$).

| Model | Dice | NSD | HD95 (mm) |
|---|---|---|---|
| **General supervised segmentation models** | | | |
| 3D U-Net | $0.1176 \pm 0.1449$ | $0.0348 \pm 0.0393$ | $78.28 \pm 40.10$ |
| SegResNet | $0.2070 \pm 0.1872$ | $0.0693 \pm 0.0649$ | $66.51 \pm 39.61$ |
| SwinUNETR | $0.1805 \pm 0.1943$ | $0.0640 \pm 0.0687$ | $65.78 \pm 29.60$ |
| SegMamba | $0.0827 \pm 0.1509$ | $0.0460 \pm 0.0885$ | $82.10 \pm 41.33$ |
| U-Mamba Bot | $0.2509 \pm 0.2480$ | $0.1834 \pm 0.1741$ | $69.62 \pm 51.43$ |
| U-Mamba Enc | $0.1559 \pm 0.2194$ | $0.1110 \pm 0.1513$ | $84.29 \pm 43.07$ |
| nnU-Net | $0.2538 \pm 0.2377$ | $0.3179 \pm 0.2106$ | $230.89 \pm 225.07$ |
| **Domain generalization variants (nnU-Net backbone)** | | | |
| GroupDRO | $0.1975 \pm 0.2242$ | $0.2886 \pm 0.1934$ | $269.99 \pm 241.09$ |
| IBERM | $0.1488 \pm 0.1941$ | $0.2280 \pm 0.1865$ | $304.35 \pm 186.50$ |
| RandConv† | $0.3050 \pm 0.2492$ | $0.3456 \pm 0.1927$ | $221.83 \pm 232.98$ |
| SD | $0.1606 \pm 0.1875$ | $0.2451 \pm 0.1859$ | $259.44 \pm 184.85$ |
| VREX | $0.2216 \pm 0.2201$ | $0.3086 \pm 0.1913$ | $227.89 \pm 213.92$ |
| **Foundation models** | | | |
| **MedSAM2** | | | |
|     Zero-shot | $0.5726 \pm 0.1931$ | $0.6689 \pm 0.1744$ | $37.86 \pm 29.97$ |
|     Full fine-tuning | $0.3950 \pm 0.1621$ | $0.5368 \pm 0.1938$ | $52.78 \pm 23.68$ |
|     Decoder-only fine-tuning | $0.3415 \pm 0.2103$ | $0.4301 \pm 0.2247$ | $72.24 \pm 43.41$ |
| **SAM-Med3D** | | | |
|     Zero-shot | $0.0799 \pm 0.0932$ | $0.0497 \pm 0.0344$ | $79.95 \pm 26.26$ |
|     Full fine-tuning | $0.3058 \pm 0.1894$ | $0.1206 \pm 0.0579$ | $38.90 \pm 23.87$ |
|     Decoder-only fine-tuning | $0.3723 \pm 0.1319$ | $0.1175 \pm 0.0352$ | $26.72 \pm 10.18$ |
| **SAM-Med3D Turbo** | | | |
|     Zero-shot | $0.1419 \pm 0.1582$ | $0.0800 \pm 0.0431$ | $50.52 \pm 26.85$ |
|     Full fine-tuning | $0.4351 \pm 0.1719$ | $0.1614 \pm 0.0537$ | $21.49 \pm 14.50$ |
|     Decoder-only fine-tuning | $0.4278 \pm 0.1302$ | $0.1407 \pm 0.0393$ | $21.50 \pm 13.16$ |
| **TotalSegmentator** | | | |
|     Zero-shot | $0.1518 \pm 0.1264$ | $0.2862 \pm 0.1595$ | $220.47 \pm 110.80$ |
|     Full fine-tuning | $0.2272 \pm 0.1938$ | $0.2721 \pm 0.1556$ | $225.85 \pm 197.87$ |
|     Decoder-only fine-tuning | $0.3352 \pm 0.2309$ | $0.3488 \pm 0.1723$ | $212.06 \pm 232.02$ |

## Appendix F. Leave-One-Sequence-Out (LOSO)

We provide the full LOSO results in Table 17 and Table 18. Table 17 reports the setting where T1W is held out (train on T2W + OOP, test on T1W), and Table 18 reports the setting where OOP is held out (train on T1W + T2W, test on OOP). These tables complement the LOSO results in the main text.

Table 17: Leave-one-sequence-out benchmark (Train on T2W+OOP $\to$ Test on T1W).

| Model | Dice | NSD | HD95 (mm) |
|---|---|---|---|
| **General supervised segmentation models** | | | |
| 3D U-Net | $0.1530 \pm 0.2182$ | $0.0505 \pm 0.0716$ | $71.15 \pm 36.15$ |
| SegResNet | $0.2338 \pm 0.2505$ | $0.0850 \pm 0.0931$ | $59.72 \pm 45.26$ |
| SwinUNETR | $0.1310 \pm 0.1867$ | $0.0500 \pm 0.0690$ | $68.80 \pm 32.04$ |
| SegMamba | $0.1610 \pm 0.1867$ | $0.0686 \pm 0.0840$ | $64.65 \pm 31.57$ |
| U-Mamba Bot | $0.3652 \pm 0.2081$ | $0.1788 \pm 0.0945$ | $67.47 \pm 42.56$ |
| U-Mamba Enc | $0.2430 \pm 0.1946$ | $0.1249 \pm 0.0734$ | $71.29 \pm 31.52$ |
| nnU-Net | $0.3258 \pm 0.2322$ | $0.2343 \pm 0.1525$ | $113.63 \pm 119.93$ |
| **Domain generalization variants (nnU-Net backbone)** | | | |
| GroupDRO | $0.4985 \pm 0.2110$ | $0.3278 \pm 0.1442$ | $90.30 \pm 126.44$ |
| IBERM | $0.4105 \pm 0.2275$ | $0.2913 \pm 0.1446$ | $125.16 \pm 134.70$ |
| RandConv | $0.5537 \pm 0.2183$ | $0.3641 \pm 0.1471$ | $92.40 \pm 135.59$ |
| SD | $0.4738 \pm 0.2355$ | $0.3271 \pm 0.1515$ | $112.68 \pm 132.22$ |
| VREX | $0.4852 \pm 0.2115$ | $0.3218 \pm 0.1457$ | $91.70 \pm 128.22$ |
| **Foundation models** | | | |
| MedSAM2 | $0.5966 \pm 0.1710$ | $0.6749 \pm 0.1646$ | $34.32 \pm 23.31$ |
| SAM-Med3D | $0.1431 \pm 0.1240$ | $0.0696 \pm 0.0431$ | $75.95 \pm 29.14$ |
| SAM-Med3D turbo | $0.4919 \pm 0.1752$ | $0.1949 \pm 0.0756$ | $26.62 \pm 17.66$ |
| TotalSegmentator | $0.4743 \pm 0.1957$ | $0.3433 \pm 0.1308$ | $89.35 \pm 115.24$ |

Table 18: Leave-one-sequence-out benchmark (Train on T1W+T2W $\to$ Test on OOP).

| Model | Dice | NSD | HD95 (mm) |
|---|---|---|---|
| **General supervised segmentation models** | | | |
| 3D U-Net | $0.4455 \pm 0.2129$ | $0.1246 \pm 0.0749$ | $27.39 \pm 17.68$ |
| SegResNet | $0.5723 \pm 0.2031$ | $0.1798 \pm 0.0915$ | $22.87 \pm 17.10$ |
| SwinUNETR | $0.3792 \pm 0.2244$ | $0.1140 \pm 0.0817$ | $38.05 \pm 22.73$ |
| SegMamba | $0.4716 \pm 0.2319$ | $0.2503 \pm 0.1533$ | $28.55 \pm 18.53$ |
| U-Mamba Bot | $0.3905 \pm 0.2404$ | $0.2301 \pm 0.1800$ | $68.53 \pm 41.65$ |
| U-Mamba Enc | $0.5197 \pm 0.2026$ | $0.3201 \pm 0.1546$ | $49.19 \pm 34.86$ |
| nnU-Net | $0.5991 \pm 0.1908$ | $0.4857 \pm 0.2029$ | $59.42 \pm 80.23$ |
| **Domain generalization variants (nnU-Net backbone)** | | | |
| GroupDRO | $0.5987 \pm 0.1923$ | $0.4907 \pm 0.2384$ | $57.09 \pm 77.97$ |
| IBERM | $0.4684 \pm 0.2254$ | $0.4023 \pm 0.2548$ | $83.81 \pm 87.42$ |
| RandConv | $0.6718 \pm 0.1833$ | $0.5565 \pm 0.2215$ | $40.51 \pm 54.59$ |
| SD | $0.5895 \pm 0.1964$ | $0.4800 \pm 0.2438$ | $75.97 \pm 108.79$ |
| VREX | $0.6318 \pm 0.1805$ | $0.5068 \pm 0.2354$ | $47.92 \pm 57.17$ |
| **Foundation models** | | | |
| MedSAM2 | $0.4751 \pm 0.1881$ | $0.5846 \pm 0.1780$ | $45.15 \pm 22.86$ |
| SAM-Med3D | $0.0935 \pm 0.0952$ | $0.0549 \pm 0.0337$ | $75.49 \pm 28.67$ |
| SAM-Med3D turbo | $0.3800 \pm 0.1595$ | $0.1332 \pm 0.0572$ | $35.59 \pm 19.53$ |
| TotalSegmentator | $0.2916 \pm 0.1654$ | $0.3187 \pm 0.1832$ | $165.54 \pm 127.22$ |

## Appendix G. Qualitative Results

To complement the primary qualitative example provided in the main text, Figure 7, Figure 8, and Figure 9 present additional cross-sequence visualization results.

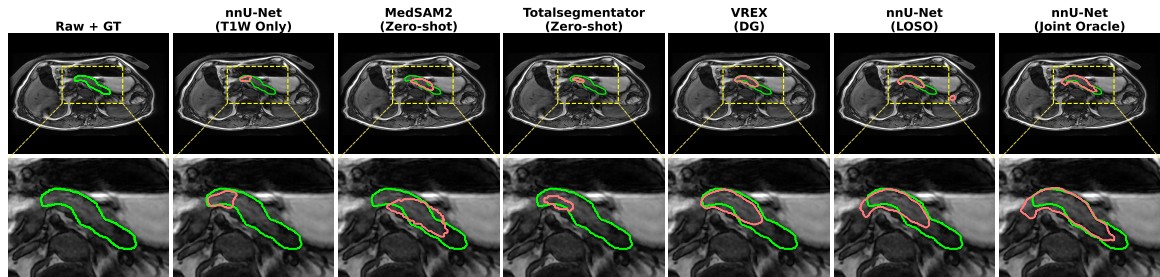

Figure 7: Qualitative comparison of pancreas MRI segmentation under the T1W → OOP sequence shift. Columns show (left → right): ground-truth annotation, nnU-Net trained only on T1W (single-source baseline), zero-shot MedSAM2, zero-shot Totalsegmentator, VREX (DG method), nnU-Net trained via Leave-One-Sequence-Out (LOSO), and nnU-Net trained jointly on all sequences (Oracle). Models trained solely on T1W commonly miss portions of the pancreas due to fat–water cancellation effects in OOP images, while zero-shot foundation models produce coarse boundaries with inconsistent localization. Green: Ground Truth; Red: Model Prediction.

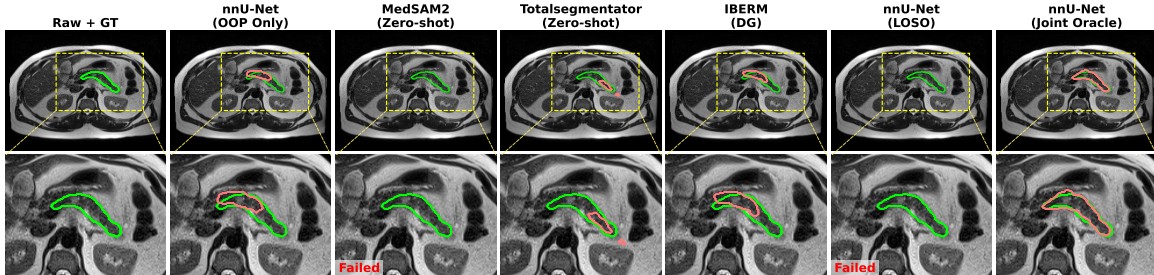

Figure 8: Qualitative comparison of pancreas MRI segmentation under the OOP → T2W sequence shift. Columns show (left → right): ground-truth annotation, nnU-Net trained only on OOP (single-source baseline), zero-shot MedSAM2, zero-shot Totalsegmentator, IBERM (DG method), nnU-Net trained via Leave-One-Sequence-Out (LOSO), and nnU-Net trained jointly on all sequences (Oracle). T2W's strong fluid contrast amplifies errors made by models trained on the weak-contrast OOP domain, causing over-segmentation or fragmented masks. Even foundation models occasionally fail under this situation. Green: Ground Truth; Red: Model Prediction.

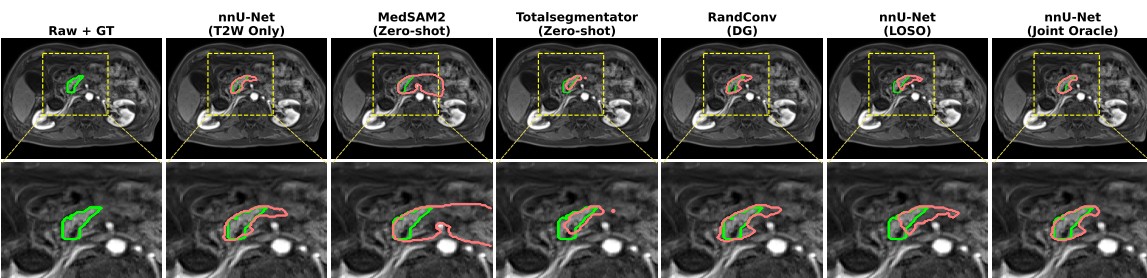

Figure 9: Qualitative comparison of pancreas MRI segmentation under the T2W → T1W sequence shift. Columns show (left → right): ground-truth annotation, nnU-Net trained only on T2W (single-source baseline), zero-shot MedSAM2, zero-shot Totalsegmentator, RandConv (DG method), nnU-Net trained via Leave-One-Sequence-Out (LOSO), and nnU-Net trained jointly on all sequences (Oracle). When evaluated on T1W, models trained on T2W often produce false positives due to the inversion of pancreas–fat intensity relations. DG methods also show instability under this strong contrast reversal. Green: Ground Truth; Red: Model Prediction.

## Appendix H. Additional Comparisons

We included two additional recent interactive models for comparison, namely VISTA3D (He et al., 2025) and nnInteractive (Isensee et al., 2025). We reported their zero-shot performances on all three sequences.

Table 19: In-domain Dice performance across T1W, T2W, and OOP MRI sequences. Foundation models are reported in zero-shot mode to highlight their intrinsic transferability, whereas other models are trained on CrossPan.

| Model | T1W Dice | T2W Dice | OOP Dice |
|---|---|---|---|
| **General supervised segmentation models** | | | |
| 3D U-Net | $0.6606 \pm 0.2901$ | $0.8078 \pm 0.0997$ | $0.6729 \pm 0.1596$ |
| SegResNet | $0.6870 \pm 0.2974$ | $0.8421 \pm 0.0755$ | $0.7338 \pm 0.1517$ |
| SwinUNETR | $0.6685 \pm 0.2921$ | $0.8168 \pm 0.0998$ | $0.7040 \pm 0.1645$ |
| SegMamba | $0.8325 \pm 0.0760$ | $0.8298 \pm 0.0875$ | $0.7305 \pm 0.1791$ |
| U-Mamba Bot | $0.8380 \pm 0.0749$ | $0.8736 \pm 0.0646$ | $0.7977 \pm 0.1474$ |
| U-Mamba Enc | $0.8435 \pm 0.0725$ | $0.8537 \pm 0.0810$ | $0.7737 \pm 0.1543$ |
| nnU-Net | $0.8353 \pm 0.0927$ | $0.8633 \pm 0.0768$ | $0.7944 \pm 0.1474$ |
| **Domain generalization variants (nnU-Net backbone)** | | | |
| GroupDRO | $0.8565 \pm 0.0669$ | $0.8764 \pm 0.0602$ | $0.7929 \pm 0.1525$ |
| IBERM | $0.8508 \pm 0.0692$ | $0.8640 \pm 0.0605$ | $0.7898 \pm 0.1473$ |
| RandConv | $0.8571 \pm 0.0652$ | $0.8743 \pm 0.0637$ | $0.7984 \pm 0.1465$ |
| SD | $0.8557 \pm 0.0664$ | $0.8754 \pm 0.0625$ | $0.7953 \pm 0.1502$ |
| VREX | $0.8548 \pm 0.0666$ | $0.8770 \pm 0.0599$ | $0.7905 \pm 0.1527$ |
| **Foundation models** | | | |
| MedSAM2 | $0.5966 \pm 0.1710$ | $0.5726 \pm 0.1931$ | $0.4751 \pm 0.1881$ |
| SAM-Med3D | $0.1431 \pm 0.1239$ | $0.0799 \pm 0.0932$ | $0.0935 \pm 0.0952$ |
| SAM-Med3D turbo | $0.4920 \pm 0.1752$ | $0.1419 \pm 0.1582$ | $0.3800 \pm 0.1596$ |
| TotalSegmentator | $0.4744 \pm 0.1958$ | $0.1518 \pm 0.1264$ | $0.2916 \pm 0.1654$ |
| VISTA3D | $0.7478 \pm 0.2833$ | $0.8461 \pm 0.0987$ | $0.6742 \pm 0.1544$ |
| nnInteractive | $0.7790 \pm 0.0957$ | $0.6602 \pm 0.1420$ | $0.7656 \pm 0.0609$ |

