# OpenReview forum: "CrossPan: A Comprehensive Benchmark for Cross-Sequence Pancreas MRI Segmentation and Generalization"
_MIDL.io/2026/Validation_Papers — MIDL 2026 - Validation Papers Poster_

### Official Review · Reviewer_qPqi · 2025-12-29

**Confidence:** 4
**Preliminary Rating:** 4
**Final Rating:** 4

**Summary:**

This paper introduces CrossPan, a multi-institutional pancreas MRI benchmark comprising 1386 annotated 3D volumes across three routinely acquired sequences (T1W, T2W, and Out-of-Phase) from 8 centers. Based on the data, the authors investigate the performance degradation when models trained on one MRI sequence are applied to another. Supervised methods, domain generalization methods, semi-supervised learning methods, and foundation models are included in the experiments, demonstrating that cross-sequence variability is more severe than cross-center variability.

**Strengths:**

1. The paper addresses a highly practical issue that is often overlooked in medical image segmentation. The manuscript displays some cases for why this problem deserves investigation. The motivation is clear.

2. The dataset is from multiple centers and covers three MRI sequences with fundamentally different contrast mechanisms, disentangling sequence-induced variability from center-induced variability, which is a key contribution.

**Weaknesses:**

1. While the benchmark and analysis are strong, the paper does not include some latest segmentation models, such as VISTA3D and nnInteractive.

2. The validation protocols for SAM-based models are not detailed. As is known, segmentation models like MedSAM2 and SAM-Med3D are promotable. It is unclear if the authors provide these models with a proper prompt. More comparisons should be done on different types of prompts (boxes or points or multiple points), as well as different protocols to simulate the user interaction.

3. The paper reports that MedSAM2 maintains moderate zero-shot performance while SAM-Med3D does not, but the underlying reasons are only briefly discussed. A more detailed analysis would strengthen the conclusions.

**Detailed Comments:**

1. Ensure consistent use of terms such as “Out-of-Phase,” “OOP,” and “chemical-shift imaging” throughout the paper to avoid confusion.

2. The compared methods could be better grouped. "General-purpose foundation model" only includes TotalSegmentator. However, SAM-based models are also "General purpose". Perhaps it would be better not to group the methods based on if they are "SAM-based".

**Justification Of Final Rating:**

The authors have addressed most of the concerns. Overall, this is a strong benchmarking study that highlights a critical limitation of current medical image segmentation models. The dataset, experiments, and clarity of motivation and conclusions make this paper a valuable contribution with high relevance to the research community. I will keep my positive score.

**Justification Of The Preliminary Rating:**

Overall, this is a strong benchmarking study that highlights a critical limitation of current medical image segmentation models. The dataset, experiments, and clarity of motivation and conclusions make this paper a valuable contribution with high relevance to the research community.

**Questions To Address In The Rebuttal:**

How are the user interaction or prompts simulated?

---

### Official Review · Reviewer_8m6A · 2026-01-09

**Confidence:** 5
**Preliminary Rating:** 3
**Final Rating:** 4

**Summary:**

This paper introduces CrossPan, a large multi-institutional benchmark for pancreas MRI segmentation, with a specific focus on evaluating cross-sequence generalization across T1W, T2W, and Out-of-Phase MRI. The study systematically compares a wide range of supervised, domain generalization, semi-supervised, and foundation models under controlled experimental settings.

**Strengths:**

- The paper highlights that cross-sequence variability can be substantially more disruptive than cross-center variability, which is an important and yet underexplored issue in abdominal MRI.

- The reported results consistently demonstrate catastrophic performance degradation under single-source cross-sequence transfer.

- Large, diverse dataset with clear splits.

**Weaknesses:**

- Given the use of multi-center clinical MRI data and the presentation of representative cases, a brief discussion of patient de-identification and privacy safeguards would be helpful.

- Translational implications could be more explicitly evaluated.

- The qualitative examples are helpful but very limited.

**Detailed Comments:**

- The validation statement would benefit from clearer demonstration of what constitutes a “validated” model under this benchmark and how such validation should inform deployment decisions.

- The comparison across many methods is valuable, but briefly contextualizing why certain models are treated as reference baselines would improve interpretability.

**Justification Of Final Rating:**

The rebuttal satisfactorily addresses the reviewer’s concerns and clearly positions the work as a validation and pre-deployment benchmarking framework. While the study is careful and well-motivated, the evidence remains largely retrospective, with limited qualitative depth in the main text and no prospective or outcome-linked validation.

**Justification Of The Preliminary Rating:**

The paper is ambitious in scope and provides a detailed empirical characterization of how sequence-induced domain shifts affect segmentation performance. Howver, the paper would benefit from clearer articulation of its validation scope, more explicit discussion of ethical considerations, and stronger connection to translational deployment.

**Questions To Address In The Rebuttal:**

Please check the Weakness and Detailed Comments sections.

---

### Official Review · Reviewer_Tovi · 2026-01-09

**Confidence:** 4
**Preliminary Rating:** 4
**Final Rating:** 4

**Summary:**

The authors introduce CrossPan, a benchmark designed to evaluate cross-sequence generalization in pancreas MRI segmentation. The dataset comprises 1,386 annotated 3D volumes spanning three clinically routine sequences (T1W, T2W, and Out-of-Phase) collected from eight international institutions. The benchmark evaluates over 15 methods, including supervised architectures, domain generalization (DG) algorithms, and foundation models.

**Strengths:**

1. The paper addresses a clinically relevant gap in medical imaging: the lack of evaluation of cross-sequence generalization in pancreas MRI segmentation.

2. The scale of the dataset (1,386 scans) is impressive for pancreas. The inclusion of multi-center data from eight institutions provides strong reliability for the benchmark.

3. The experimental design is logically sound and comprehensive for existing domain generalization methods.

**Weaknesses:**

1. The specific contributions in the Introduction could be articulated more clearly. For instance, the authors should explicitly state whether the curated CrossPan dataset itself is intended as a contribution.
2. The findings are currently based on the internal CrossPan benchmark. It remains unclear whether the observations across sequences is reproducible on other external, publicly available datasets.
3. The manuscript lacks a detailed description of the cohort selection process. It is unclear if cases were selected based on specific criteria that might introduce bias.
4. Several experimental details are missing. For instance, the prompt strategies used for training and testing of SAM-based models are not clear. And it is not clear whether the baseline models were trained from scratch or initialized with official pretrained weights.

**Detailed Comments:**

In Table 1, which models are zero-shot results and which models are trained on CrossPan? If the SAM-based models are evaluated only in zero-shot mode, the comparison with fully supervised models trained on CrossPan may be not fair enough. I recommend the authors include fine-tuned versions of these foundation models here.

**Justification Of Final Rating:**

The authors have provided clear and precise responses to my questions. I have no remaining concerns. CrossPan is valuable to the MIDL community, and I would maintain my rating of 4. I support its publication at MIDL.

**Justification Of The Preliminary Rating:**

The paper provides a significant contribution by establishing CrossPan, a large-scale (1,386 scans) and multi-institutional benchmark. The rigor of the evaluation across 15+ models and the scale of the data make it valuable to the MIDL community. Addressing the requested clarifications and specific implementation details would further strengthen the work.

**Questions To Address In The Rebuttal:**

1. Could the authors provide external validation on other public datasets (e.g., AMOS/BTCV) that contain pancreas MRI data?
2. While the paper focuses on T1W to T2W transfer, could the authors provide extra analysis on the reverse transfer (e.g., training on T2W and testing on T1W) and explore the model performance in this case?
3. What is the details of data construction? Authors mentioned that data is extended from their previous work. What is the difference between two versions?
4. More experiment details are expected, e.g. model initialization (scratch vs. checkpoint) and the specific prompt used for the SAM family models.

---

### Author Rebuttal · Authors · 2026-01-24

**Rebuttal:**

We thank the reviewers for their constructive feedback. Based on their suggestions, we have revised the manuscript to include new model comparisons and external validation, and clarify the contributions, data privacy, and cohort selection. We have also strengthened the discussion on deployment implications and foundation model analysis. All changes are highlighted in the revised manuscript.

**Supporting Material:**

/attachment/45bbe62146c6c536d70abc775a935b37a357276a.pdf

---

### Meta-Review · Area_Chair_ZH5c · 2026-02-03

**Recommendation:** Accept (Oral)
**Confidence:** 5

**Metareview:**

All reviewers are postive to this paper and acknowledge the value of this benchmarking work.

---

### Decision · Program_Chairs · 2026-02-14

Accept (Poster)